# RaGEP: Rank-aware Geometric Expert Pruning for Mixture-of-Experts Language Models

Wentao Hu [* 1]  Zeyu Zhu [* 2]  Mingkuan Zhao [1]  Zhenhua An [1]  Yanbo Zhai [1]  Shanhong Yu [3]  Huilin Zhou [4]  Xin Lai [1]  Xiaoyan Zhu [1]  Jiayin Wang [1]

## Abstract

Sparse Mixture-of-Experts (MoE) architectures scale model capacity efficiently but suffer from massive static parameter footprints, creating significant deployment burdens on memory-constrained hardware. Existing post-training pruning methods often rely on scalar statistics, ignoring the representational geometry of expert feature spaces. This leads to sub-optimal resource allocation across layers and the retention of redundant experts. To address this, we propose a Rank-aware Geometric Expert Pruning (RaGEP) framework to compress MoE models by analyzing the geometric properties of expert activations. First, in the inter-layer allocation stage, we introduce a Rank-aware budget allocation mechanism that adaptively assigns expert budgets based on the effective rank of layer-wise representations. Second, in the intra-layer selection stage, we propose a Spectral-Salience Pruning metric that harmonizes subspace orthogonality and activation magnitude to identify high-energy orthogonal experts. Extensive experiments across MoE models of different scales show that our method consistently outperforms state-of-the-art baselines on a diverse set of zero-shot tasks, while reducing model size and inference cost. Code is available at `https://github.com/Saul-James/RaGEP`.

## 1. Introduction

The rapid evolution of Large Language Models (LLMs) has fundamentally reshaped the landscape of natural language processing (Yang et al., 2025; OpenAI et al., 2024). To scale model capacity while maintaining manageable inference costs, the Sparse Mixture-of-Experts (MoE) architecture has emerged as a dominant paradigm. By activating only a small subset of parameters per token, MoE models decouple computational load from parameter count to achieve superior inference efficiency (Jiang et al., 2024; Yang et al., 2025; Liu et al., 2024). Despite sparse per-token activation, deployment remains constrained by the massive static expert pool. In practice, systems must load tens to hundreds of experts, resulting in prohibitive VRAM footprints, high initialization latency, and significant inter-device communication overhead. To this end, post-training expert pruning serves as a compression strategy that is orthogonal and complementary to weight quantization. By directly eliminating redundant experts at the structural level rather than merely reducing numerical precision, it acts as a critical bridge connecting high-performance MoE models with limited hardware resources.

While existing pruning paradigms have made strides, their design philosophy often suffers from a fundamental geometric misalignment. Early works relied on task-specific fine-tuning (Chen et al., 2022). Recent retraining-free approaches, such as Enumeration-based Expert Pruning (Lu et al., 2024), Mosaic Pruning (Hu et al., 2026b), and the hierarchical clustering-based merging method HC-SMoE (Chen et al., 2025), primarily select or merge experts based on scalar statistics like activation frequency, reconstruction loss, or output similarity. These methods treat the MoE model as a collection of independent modules and neglect the deep representational geometry of the expert feature spaces, which can lead to removing complementary experts or retaining redundant ones, ultimately degrading quality–compression trade-offs, and often precipitating a catastrophic functional collapse that renders the model ineffective on complex reasoning tasks.

On one hand, regarding inter-layer allocation, current uniform pruning strategies operate under the flawed assumption that all layers contribute equally to model capacity. This ignores the fact that different layers construct feature subspaces with drastically different effective ranks. Con-

*Equal contribution  Contact email: wentao_hu@stu.xjtu.edu.cn. [1]Xi'an Jiaotong University, Xi'an, China [2]Institute of Automation, Chinese Academy of Sciences, Beijing, China [3]Beijing Foreign Studies University, Beijing, China [4]University of Science and Technology of China, Hefei, China. Correspondence to: Jiayin Wang <wangjiayin@mail.xjtu.edu.cn>.

*Proceedings of the 43rd International Conference on Machine Learning*, Seoul, South Korea. PMLR 306, 2026. Copyright 2026 by the author(s).

sequently, such strategies risk destroying high-density information in critical layers while retaining redundancy in low-rank layers. On the other hand, regarding intra-layer selection, methods prioritizing reconstruction loss (Lu et al., 2024) or output similarity tend to retain experts that are popular or statistically dominant. In over-parameterized models, multiple experts often converge to similar feature subspaces and create linear redundancy. Retaining such high-activation but functionally collinear experts is inefficient. Conversely, diversity-focused methods (Hu et al., 2026b) often overlook activation energy and potentially retain noise experts that possess unique directions but make negligible contributions to the residual stream.

To address these challenges, we propose a Rank-aware Geometric Expert Pruning framework (RaGEP), which is retraining-free and systematically compresses MoE models by utilizing the geometric properties of experts. Unlike previous approaches that rely on heuristics, RaGEP first introduces Rank-aware Budget Allocation (RBA), a mechanism that adaptively assigns expert budgets based on the effective rank of layer-wise representations. By quantifying the intrinsic dimensionality of each layer's covariance matrix via effective rank, we adaptively allocate higher expert budgets to high-rank layers to preserve complex feature structures while aggressively compressing low-rank layers. Furthermore, for expert selection within layers, we formulate the problem as a search for a high-energy orthogonal basis. We propose Spectral-Salience Pruning (SSP), a unified metric that harmonizes spectral novelty and activation salience. This fusion allows RaGEP to effectively filter out redundant experts with low novelty and silent experts with low energy, which ensures the retained experts are both functionally distinct and highly influential.

Our main contributions are summarized as follows:

- We identify the geometric misalignment in existing MoE compression methods, including pruning and clustering-based merging, and demonstrate that scalar statistics are insufficient for detecting linear redundancy in expert feature spaces.

- We propose RaGEP, a retraining-free framework that incorporates Rank-aware Budget Allocation to adaptively distribute sparsity based on information density, and Spectral-Salience Pruning to select a minimal sufficient basis of experts.

- We conduct extensive experiments on representative large-scale MoE models, including Qwen3-30B-A3B, Mixtral 8x7B, and DeepSeek-V2-Lite (16B). The results across diverse benchmarks demonstrate that our method significantly outperforms state-of-the-art baselines while establishing a new standard for structural compression robustness.

**Conflict of Interest Disclosure.** The authors declare no financial conflicts of interest related to this work. All evaluated models (Qwen3-30B-A3B, Mixtral 8x7B, and DeepSeek-V2-Lite) are publicly released open-source models, and none of the authors are affiliated with the organizations that developed them.

## 2. Related Work

The Mixture-of-Experts (MoE) architecture scales capacity by activating sparse parameter subsets, decoupling computational cost from model size (Shazeer et al., 2017; Fedus et al., 2022). Recent advances adapted this paradigm to decoder-only LLMs, yielding models such as Mixtral 8x7B (Jiang et al., 2024), Qwen3-30B-A3B (Yang et al., 2025), and DeepSeek-V2-Lite (Liu et al., 2024). While achieving superior inference efficiency, the necessity to load all experts creates significant deployment bottlenecks on resource-constrained hardware due to the high static memory footprint. While achieving superior inference efficiency, the necessity to load all experts creates significant deployment bottlenecks on resource- constrained hardware due to the high static memory footprint. Complementary directions tackle MoE efficiency from orthogonal angles: at the attention level, principled structural sparsity reorganizes the computational workload across heads to eliminate redundancy (Zhao et al., 2026); at the routing level, counterfactual analysis identifies and awakens dormant specialist experts to mitigate hallucinations on long-tail knowledge (Hu et al., 2026a). Our work focuses on alleviating this structural redundancy through efficient post-training compression. Our work focuses on alleviating this structural redundancy through efficient post-training compression.

### 2.1. Post-training Expert Pruning

Expert pruning aims to permanently remove redundant experts to reduce the static parameter footprint. Early approaches relied on task-specific fine-tuning (Chen et al., 2022) or dynamic skipping mechanisms (Lu et al., 2024), which often incur high training costs or fail to reduce static memory usage. Recent retraining-free strategies, such as Enumeration-based Expert Pruning (Lu et al., 2024), Mosaic Pruning (Hu et al., 2026b), and HC-SMoE (Chen et al., 2025), primarily select or merge experts based on scalar metrics like reconstruction loss, functional diversity, or output similarity. However, these methods largely rely on scalar statistics, such as activation frequency, loss reduction, or Euclidean distance between outputs. They treat experts as independent units and often fail to detect linear redundancy where experts span overlapping subspaces. In contrast, RaGEP adopts a geometric perspective by evaluating experts based on the orthogonality (spectral novelty) and magnitude (energy) of their activation subspaces, thus iden-

tifying redundancy at a deeper representational level.

## 2.2. Geometric Analysis of Representations

Understanding the geometry of internal representations is crucial for model compression. Chi et al. (2022) observed the phenomenon of representation collapse in MoEs, where different experts converge to similar feature subspaces and lead to representational redundancy. Furthermore, representation analysis in dense LLMs has shown that different layers exhibit varying degrees of anisotropy and intrinsic dimensionality (Ethayarajh, 2019). Building on these insights, our work is the first to explicitly leverage Effective Rank to quantify layer-wise information density for adaptive budget allocation, and to utilize subspace projections to resolve expert-level linear redundancy.

## 3. Methodology

### 3.1. Problem Formulation

Consider a Transformer-based MoE layer $l$ within the RaGEP framework (Figure 1), with input $\mathbf{x} \in \mathbb{R}^d$, where $d$ denotes the hidden dimension, experts $\mathcal{E}^{(l)}$ and router $G$. The output of the full layer is given by $\mathbf{y}^{(l)} = \sum_{i=1}^{N} G(\mathbf{x})_i \cdot E_i^{(l)}(\mathbf{x})$. To formulate the pruning objective, we define the output of a pruned subset $\mathcal{S}^{(l)} \subset \mathcal{E}^{(l)}$ as:

$$\hat{\mathbf{y}}^{(l)}(\mathcal{S}^{(l)}) = \sum_{E_i \in \mathcal{S}^{(l)}} G(\mathbf{x})_i E_i^{(l)}(\mathbf{x}). \quad (1)$$

We select subsets $\{\mathcal{S}^{(l)}\}_{l=1}^{L}$ subject to budgets $B^{(l)}$ (number of retained experts) that minimize the expected reconstruction error over the calibration distribution $\mathcal{D}$:

$$\min_{\{\mathcal{S}^{(l)}\}_{l=1}^{L}} \sum_{l=1}^{L} \mathbb{E}_{\mathbf{x} \sim \mathcal{D}} \left[ \left\| \mathbf{y}^{(l)} - \hat{\mathbf{y}}^{(l)}(\mathcal{S}^{(l)}) \right\|_2^2 \right]. \quad (2)$$

Directly optimizing Eq.(2) is computationally intractable and tends to bias selection towards generalist experts, often neglecting specialists crucial for specific tasks. To address this, RaGEP moves beyond simple reconstruction approximation; it identifies a high-energy orthogonal basis via geometric analysis, introducing a structural inductive bias that balances representational fidelity with functional diversity.

### 3.2. Rank-aware Budget Allocation via Effective Rank

Previous pruning methods typically enforce a uniform budget $B^{(l)} = B$ across all layers, implicitly assuming that every layer contributes equally to the model's capacity. However, our empirical analysis of pre-trained MoE models reveals a significant topological heterogeneity in their feature spaces.

As illustrated in Figure 2, the effective rank ($r_l$)—a proxy for the intrinsic dimensionality of the feature space—varies

drastically across layers. For instance, Qwen3-30B-A3B exhibits extremely low ranks ($r_l < 10$) in early layers, indicating severe feature collapse and high redundancy, whereas middle layers sustain high information density. Similarly, Mixtral 8x7B displays a distinct bell-shaped rank distribution. A uniform pruning strategy implies a geometric misalignment: it would drastically under-parameterize high-rank layers (destroying critical information) while over-allocating budget to low-rank layers (preserving representational redundancy where a minimal basis would suffice).

To address this, RaGEP explicitly quantifies this information density to guide resource allocation.

**Spectral Entropy and Information Density.** Let $\mathbf{H}^{(l)} \in \mathbb{R}^{M \times d}$ denote the matrix of hidden states collected from calibration data ($M$ tokens). We analyze the centralized covariance matrix $\boldsymbol{\Sigma}_l \in \mathbb{R}^{d \times d}$:

$$\boldsymbol{\Sigma}_l = \frac{1}{M-1}(\mathbf{H}^{(l)} - \bar{\mathbf{H}}^{(l)})^\top (\mathbf{H}^{(l)} - \bar{\mathbf{H}}^{(l)}). \quad (3)$$

The eigenvalues $\lambda_1 \geq \lambda_2 \geq \cdots \geq \lambda_d \geq 0$ of $\boldsymbol{\Sigma}_l$ characterize the energy distribution along principal directions. We define the normalized spectral distribution $\mathbf{p} = \{p_j\}_{j=1}^{d}$ as:

$$p_j = \frac{\lambda_j}{\sum_{k=1}^{d} \lambda_k} = \frac{\lambda_j}{\text{tr}(\boldsymbol{\Sigma}_l)}, \quad \text{s.t.} \quad \sum_{j=1}^{d} p_j = 1. \quad (4)$$

The effective rank (Roy & Vetterli, 2007) is defined as the exponential of the spectral Shannon entropy $H(\mathbf{p})$:

$$\begin{aligned} \text{erank}(\boldsymbol{\Sigma}_l) &\triangleq \exp\left(H(\mathbf{p})\right) \\ &= \exp\left(-\sum_{j=1}^{d} p_j \log p_j\right). \end{aligned} \quad (5)$$

*Intuition:* The effective rank measures the volume of the feature space effectively occupied by the data. A low erank implies that the layer's activations are confined to a narrow cone in $\mathbb{R}^d$ (highly anisotropic), indicating high redundancy and low sensitivity to pruning. Conversely, a high effective rank suggests an isotropic distribution where information is spread uniformly across many dimensions, necessitating a larger expert budget to preserve representational fidelity.

**Efficient Trace Approximation.** Since full eigendecomposition involves prohibitive computational costs ($\mathcal{O}(d^3)$), we employ a stable approximation derived from the relationship between Shannon entropy and collision entropy (Rényi entropy of order 2) (Rudelson & Vershynin, 2007). To explicitly demonstrate the computational efficiency, we formulate the layer-wise approximate effective rank $r_l$ using the Frobenius norm:

$$r_l \triangleq \frac{(\text{tr}(\boldsymbol{\Sigma}_l))^2}{\text{tr}(\boldsymbol{\Sigma}_l^2)} = \frac{(\text{tr}(\boldsymbol{\Sigma}_l))^2}{\|\boldsymbol{\Sigma}_l\|_F^2}. \quad (6)$$

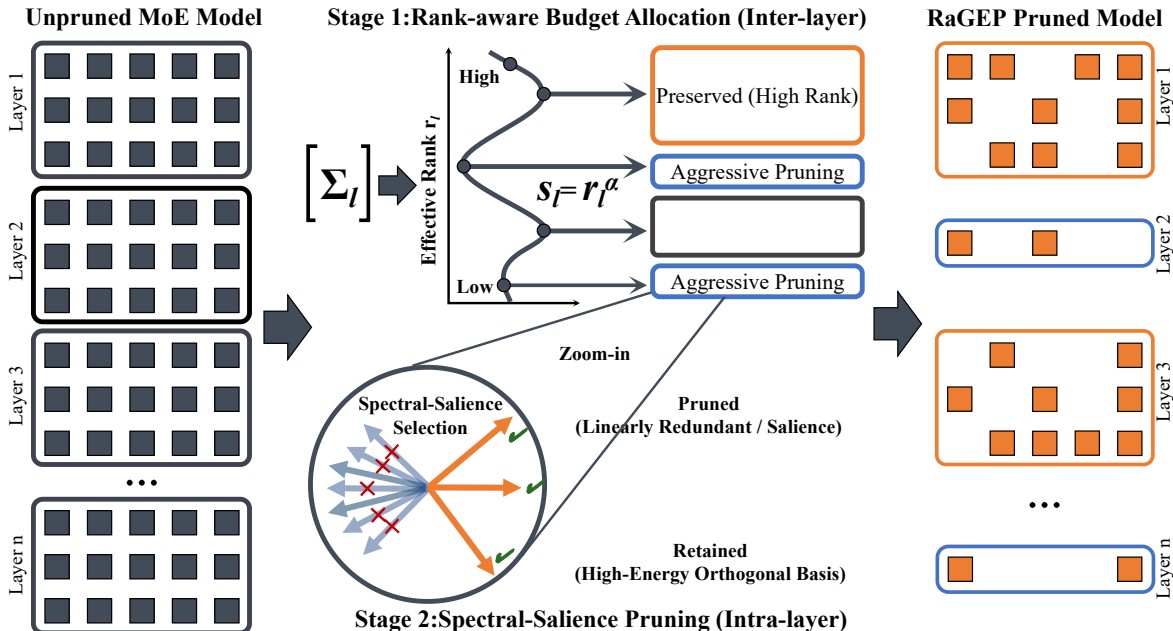

*Figure 1.* **The overall framework of RaGEP.** The process consists of two coarse-to-fine stages: (1) Rank-aware Budget Allocation (Inter-layer) adaptively determines the layer-specific sparsity budget based on the effective rank ($\mathbf{\Sigma}_l$) of feature spaces, allocating more experts to information-dense layers; (2) Spectral-Salience Pruning (Intra-layer) selects experts that form a high-energy orthogonal basis by harmonizing Spectral Novelty (geometric orthogonality) and Activation Salience (magnitude intensity).

Unlike full eigendecomposition ($\mathcal{O}(d^3)$) or explicit matrix multiplication, computing the Frobenius norm $\|\mathbf{\Sigma}_l\|_F^2$ only requires element-wise squaring and summation, strictly bounding the complexity to $\mathcal{O}(d^2)$ (excluding the $\mathcal{O}(Md^2)$ cost for covariance accumulation). Consequently, this metric serves as a robust proxy for layer sensitivity. We derive the layer-wise budget by scaling the global budget $K_{\text{total}}$ proportional to the sensitivity score $s_l = (r_l)^\alpha$:

$$B^{(l)} = \max\left(k_{\min}, \left\lfloor \frac{s_l}{\sum_{j=1}^{L} s_j} \cdot K_{\text{total}} \right\rfloor\right), \quad (7)$$

where $\alpha \geq 0$ is a hyperparameter controlling the steepness of the allocation strategy.

### 3.3. Spectral-Salience Pruning via Geometric Projection

Having determined the layer-wise budgets, the challenge shifts to selecting the optimal experts. Purely loss-based methods tend to select popular experts, which often converge to similar feature subspaces, creating linear redundancy. We propose Spectral-Salience Pruning (SSP), conceptually depicted in Figure 3, to identify a basis set that is both functionally orthogonal and energetically significant.

#### 3.3.1. SUBSPACE MODELING VIA VARIATIONAL PCA

For each expert $E_i$, we construct its local activation matrix $\mathbf{A}_i \in \mathbb{R}^{m_i \times d}$ by stacking the output vectors of the $m_i$ tokens routed to $E_i$. To capture the principal functional directions,

we perform Principal Component Analysis (PCA) to extract the top-$r$ singular vectors. This corresponds to solving the variance maximization problem on the Stiefel manifold:

$$\mathbf{U}_i = \underset{\mathbf{U} \in \mathbb{R}^{d \times r}, \mathbf{U}^\top \mathbf{U} = \mathbf{I}_r}{\arg\max} \operatorname{tr}(\mathbf{U}^\top \mathbf{A}_i^\top \mathbf{A}_i \mathbf{U}). \quad (8)$$

The column space of $\mathbf{U}_i$, denoted as $\mathcal{V}_i = \operatorname{span}(\mathbf{U}_i)$, represents the functional subspace of expert $E_i$.

#### 3.3.2. SPECTRAL NOVELTY VIA ORTHOGONAL PROJECTION

We quantify the redundancy of expert $E_i$ by measuring the overlap between its subspace $\mathcal{V}_i$ and the union of all other experts' subspaces $\mathcal{V}_{-i} = \bigcup_{j \neq i} \mathcal{V}_j$. We obtain the orthonormal basis $\mathbf{U}_{-i}$ for $\mathcal{V}_{-i}$ via QR decomposition (see Appendix D.1 for efficient implementation details). The projection of $\mathbf{U}_i$ onto $\mathcal{V}_{-i}$ is given by the idempotent projection operator $\mathbf{P}_{-i} = \mathbf{U}_{-i}\mathbf{U}_{-i}^\top$. The redundant projection strength is:

$$\begin{aligned} \rho(i) &= \|\mathbf{P}_{-i}\mathbf{U}_i\|_F^2 \\ &= \operatorname{tr}(\mathbf{U}_i^\top \mathbf{P}_{-i}^\top \mathbf{P}_{-i} \mathbf{U}_i) \quad (9) \\ &= \operatorname{tr}(\mathbf{U}_i^\top \mathbf{U}_{-i}\mathbf{U}_{-i}^\top \mathbf{U}_i). \end{aligned}$$

Here, $\rho(i) \in [0, r]$. Geometrically, this value represents the sum of squared cosines of the principal angles between subspace $\mathcal{V}_i$ and $\mathcal{V}_{-i}$. If $\rho(i) \approx r$, the functional subspace of $E_i$ is almost entirely contained within the span of other experts, rendering it linearly redundant. We thus define the

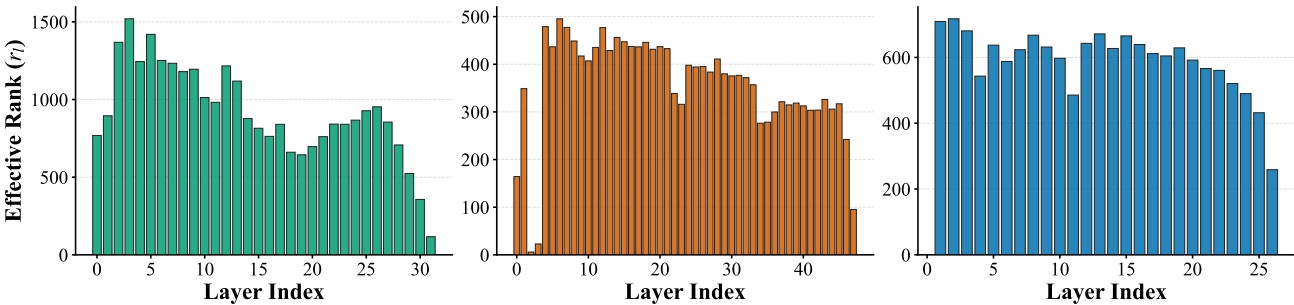

*Figure 2.* **Layer-wise Effective Rank analysis of unpruned MoE models.** We visualize the effective rank ($r_l$) of expert activations for Mixtral 8x7B (left), Qwen3-30B-A3B (middle), and DeepSeek-V2-Lite (right). The significant variation in rank values across layers highlights the necessity of our Rank-aware Budget Allocation strategy: layers with higher effective ranks (higher information density) require a larger expert budget to preserve representational capacity, while low-rank layers can be aggressively pruned without performance loss.

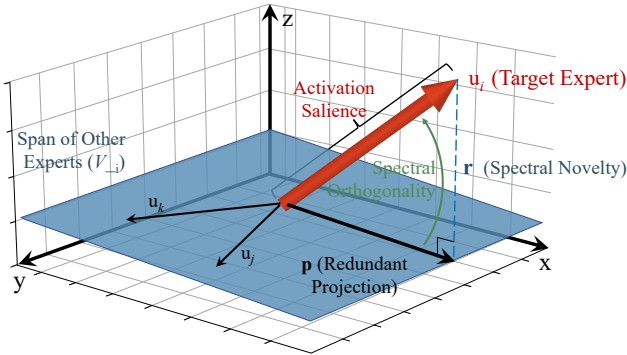

*Figure 3.* **Geometric intuition of Spectral-Salience Pruning (SSP).** The target expert vector $\mathbf{u}_i$ (Red) is decomposed relative to the subspace spanned by other experts ($V_{-i}$, Blue Plane). Spectral Novelty is quantified by the orthogonal residual component $\mathbf{r}$ representing functional uniqueness, while Activation Salience corresponds to the vector magnitude (representing contribution intensity). RaGEP selects experts that maximize both metrics simultaneously.

Spectral Novelty Score as:

$$\phi_{\text{spec}}(i) = 1 - \frac{1}{r}\rho(i). \tag{10}$$

This score prioritizes experts that contribute unique, orthogonal directions to the representational geometry.

### 3.3.3. ACTIVATION SALIENCE AND HARMONIZATION

Spectral novelty solely captures directionality. To filter out noise experts that possess unique directions but negligible magnitudes, we introduce Activation Salience, measured by the total magnitude (energy) contributed to the residual stream:

$$\phi_{\text{sal}}(i) = \|\mathbf{A}_i\|_F^2 = \sum_{\mathbf{x} \in \mathcal{D}_i} \|E_i(\mathbf{x})\|_2^2, \tag{11}$$

where $\mathcal{D}_i$ denotes the subset of tokens routed to expert $E_i$. To robustly fuse the geometric orthogonality and functional magnitude, we employ a rank-based harmonization. The final importance score $\mathcal{S}_{\text{SSP}}(i)$ is defined as:

$$\mathcal{S}_{\text{SSP}}(i) = (1 - \beta) \cdot \frac{\mathcal{R}(\phi_{\text{spec}}(i))}{N} + \beta \cdot \frac{\mathcal{R}(\phi_{\text{sal}}(i))}{N}, \tag{12}$$

where $\mathcal{R}(\cdot)$ is the ranking function that maps raw scores to integer ranks $\{1, \ldots, N\}$ in ascending order , i.e., assigning 1 to the lowest score and $N$ to the highest, and $\beta \in [0, 1]$ is a balancing factor. By selecting experts with top $\mathcal{S}_{\text{SSP}}$ scores, RaGEP effectively filters out both linearly redundant experts (high salience, low novelty) and silent experts (high novelty, low salience), ensuring the retained experts form a high-energy orthogonal basis for the model.

## 4. Experiments

### 4.1. Experimental Setup

**Models.** We conduct experiments on three representative open-source MoE models: Qwen3-30B-A3B, Mixtral 8x7B, and DeepSeek-V2-Lite. These models vary significantly in architecture and expert count, allowing us to evaluate the scalability of RaGEP across different regimes.

**Implementation Details.** All experiments were conducted on a node equipped with four NVIDIA A100 GPUs with 80GB memory. We use the C4 dataset (Dodge et al., 2021) for calibration, randomly sampling 32 sequences with a length of 2048 tokens.

**Baselines.** We compare RaGEP against three state-of-the-art post-training pruning methods. **EEP** (Enumeration-based Expert Pruning) (Lu et al., 2024) selects experts by minimizing the reconstruction loss; due to the combinatorial explosion of the search space on large-scale models (e.g., Qwen and DeepSeek), we adopt a greedy layer-wise search strategy following prior work (He et al., 2024) to ensure computational feasibility. **MoP** (Mosaic Pruning) (Hu et al., 2026b) utilizes clustering to preserve functional diversity among experts. **HC-SMoE** (Chen et al., 2025) employs

a hierarchical clustering-based merging approach; in our reproduction, to strictly enforce the parameter-matching constraint and maintain routing consistency, we applied frequency-weighted averaging to both the expert parameters and their corresponding router embeddings within each cluster.

**Datasets & Metrics.** We evaluate zero-shot performance on eight standard benchmarks: ARC-Challenge (ARC-c) and ARC-Easy (ARC-e) (Clark et al., 2018), BoolQ (Clark et al., 2019), HellaSwag (Zellers et al., 2019), MMLU (Hendrycks et al., 2021), OpenBookQA (OBQA) (Mihaylov et al., 2018), RTE (Bentivogli et al., 2009), and WinoGrande (Sakaguchi et al., 2021). We report the average accuracy across these tasks.

## 4.2. Main Results

Table 1 presents the zero-shot performance of RaGEP compared to state-of-the-art baselines at a 50% pruning ratio. Across all three model architectures, RaGEP consistently achieves superior performance, demonstrating the universality of our geometric approach. Results at a 75% retention ratio (Appendix C.1) further confirm RaGEP's superiority and robust performance with minimal degradation. On Qwen3-30B-A3B, while RaGEP maintains the highest average accuracy, its advantage is particularly decisive on knowledge-intensive benchmarks. Notably, on MMLU, RaGEP achieves a score of 58.45, outperforming the strongest baseline on this benchmark (EEP) by a significant margin of 2.78 points (58.45 vs. 55.67). This indicates that scalar-based methods tend to discard experts critical for complex semantic reasoning, whereas RaGEP's spectral selection effectively preserves these high-value components.

Furthermore, RaGEP demonstrates remarkable robustness on DeepSeek-V2-Lite, a model characterized by its fine-grained expert design which makes it highly susceptible to performance collapse under compression. In this challenging regime, baseline methods struggle significantly (e.g., EEP drops to 24.55 on MMLU), whereas RaGEP successfully mitigates functional collapse and consistently outperforms baselines across almost all tasks. Similarly, on Mixtral 8x7B, RaGEP achieves the best performance recovery relative to the unpruned model. By adaptively allocating budgets based on effective rank, RaGEP ensures that critical layers retain sufficient capacity, thereby establishing a new state-of-the-art for structural MoE compression.

## 4.3. Ablation Studies

To disentangle the individual contributions of the proposed components, we conduct a component-wise ablation study on the Qwen3-30B-A3B model at two different expert retention ratios (50% and 75%). We use the standard EEP method as our baseline, which utilizes uniform budget allocation

and loss-based expert selection. We introduce two variants for comparison: "+ RBA" denotes replacing the uniform budget with our rank-aware budget allocation while keeping the selection metric loss-based; "+ SSP" denotes keeping the uniform budget but replacing the selection metric with our Spectral-Salience Pruning. The comprehensive results are summarized in Table 2.

The results validate the individual and synergistic contributions of RaGEP's components. At 50% retention, incorporating rank-aware allocation alone improves average accuracy to 61.96, while employing SSP alone raises it to 62.18. Crucially, combining both strategies in the full RaGEP framework yields the highest average accuracy of 63.12. This indicates that adaptive resource allocation and geometric expert selection are complementary: the former ensures budgets are directed to information-dense layers, while the latter selects functionally orthogonal experts. This superiority holds consistent at the 75% retention ratio, confirming the robustness of our approach across different compression levels.

## 4.4. Sensitivity to Hyperparameter $\beta$

We analyze the sensitivity of the balancing factor $\beta$ (Eq.12) on DeepSeek-V2-Lite, varying it from $0.0$ to $1.0$ under two pruning ratios.

As illustrated in Figure 4, the model performance improves significantly as we introduce salience information, peaking at $\beta = 0.8$ before experiencing a slight decline. The steady improvement from $\beta = 0.0$ to $\beta = 0.8$ indicates that activation magnitude serves as the primary signal for expert importance. However, the slight performance drop observed when shifting from $\beta = 0.8$ to $\beta = 1.0$ (pure salience) suggests that relying solely on magnitude tends to retain linearly redundant experts. By incorporating a small weight for spectral novelty (i.e., $1 - \beta = 0.2$), RaGEP effectively filters out this redundancy without discarding high-contribution components. This hybrid configuration yields the optimal trade-off, surpassing both single-metric strategies and standard baselines. Consequently, we set $\beta = 0.8$ as the default for all subsequent experiments. Detailed numerical results are provided in Appendix C.2.

## 4.5. Robustness across Expert Retention Ratios

To evaluate the stability , we conduct a sensitivity analysis on DeepSeek-V2-Lite. This model's fine-grained expert architecture makes it a rigorous testbed for parameter removal. Figure 5 illustrates the performance trajectory.

As observed, RaGEP demonstrates high resilience across the sparsity spectrum. Notably, at the 75% retention ratio, the pruned model achieves an average accuracy of 59.20, retaining nearly 92% of the original model's performance.

*Table 1.* Zero-shot comparison of Mixtral 8x7B, DeepSeek-V2-Lite, and Qwen3-30B-A3B at a 50% pruning ratio. "None" denotes the original unpruned model. **RaGEP** achieves the best average performance across all tasks, significantly mitigating the performance drop compared to EEP and MoP. The best average results are highlighted in **bold**.

| Model | Method | ARC-c | ARC-e | BoolQ | HellaSwag | MMLU | OBQA | RTE | WinoGrande | Average |
|---|---|---|---|---|---|---|---|---|---|---|
| Mixtral | None | 62.37 | 87.08 | 88.41 | 67.61 | 68.69 | 37.20 | 71.12 | 77.11 | 69.95 |
| | HC-SMoE | 46.33 | 75.67 | 84.83 | 57.15 | 51.68 | 29.20 | 74.73 | 69.69 | 61.16 |
| | EEP | 53.50 | 80.93 | 84.68 | 60.72 | 50.61 | 31.00 | 67.87 | 73.56 | 62.86 |
| | MoP | 53.41 | 76.09 | 84.56 | 60.02 | 52.51 | 30.60 | 72.29 | 73.72 | 62.90 |
| | **RaGEP** | 52.99 | 80.30 | 84.77 | 60.34 | 51.89 | 32.20 | 72.92 | 75.22 | **63.83** |
| DeepSeek | None | 52.65 | 80.35 | 82.81 | 62.49 | 56.67 | 35.60 | 72.56 | 71.11 | 64.28 |
| | HC-SMoE | 27.05 | 48.15 | 62.39 | 37.07 | 33.21 | 17.14 | 56.68 | 56.59 | 42.29 |
| | EEP | 34.22 | 59.64 | 60.46 | 46.38 | 24.55 | 25.20 | 57.40 | 57.77 | 45.70 |
| | MoP | 32.25 | 58.59 | 64.83 | 45.61 | 26.39 | 21.80 | 61.73 | 56.51 | 45.96 |
| | **RaGEP** | 34.73 | 60.35 | 62.63 | 48.80 | 26.28 | 26.80 | 58.84 | 60.22 | **47.33** |
| Qwen | None | 52.82 | 79.62 | 86.68 | 59.46 | 77.83 | 34.00 | 82.67 | 70.16 | 67.91 |
| | HC-SMoE | 40.61 | 69.82 | 85.35 | 54.11 | 42.79 | 30.40 | 62.82 | 69.38 | 56.91 |
| | EEP | 42.92 | 72.14 | 87.40 | 56.48 | 55.67 | 30.38 | 77.62 | 70.17 | 61.60 |
| | MoP | 46.16 | 76.22 | 88.20 | 56.79 | 52.07 | 30.80 | 77.62 | 69.61 | 62.18 |
| | **RaGEP** | 46.47 | 76.14 | 88.20 | 56.82 | 58.45 | 30.40 | 78.34 | 70.17 | **63.12** |

*Table 2.* Ablation study on Qwen3-30B-A3B at 50% and 75% expert retention ratios. EEP: The baseline using uniform budget and loss-based selection. + RBA: Applies Rank-aware Budget Allocation with loss-based selection. + SSP: Applies uniform budget with Spectral-Salience Pruning. RaGEP: Combines both rank-aware budget allocation and SSP. The best average results are highlighted in **bold**.

| Setting | Method | ARC-c | ARC-e | BoolQ | HellaSwag | MMLU | OBQA | RTE | WinoGrande | Average |
|---|---|---|---|---|---|---|---|---|---|---|
| Retain 50% | EEP (Baseline) | 42.92 | 72.14 | 87.40 | 56.48 | 55.67 | 30.38 | 77.62 | 70.17 | 61.60 |
| | + RBA | 45.48 | 72.43 | 87.92 | 56.96 | 53.12 | 32.00 | 76.53 | 71.27 | 61.96 |
| | + SSP | 46.25 | 75.84 | 87.68 | 55.51 | 50.80 | 30.20 | 81.95 | 69.22 | 62.18 |
| | **RaGEP (Full)** | 46.47 | 76.14 | 88.20 | 56.82 | 58.45 | 30.40 | 78.34 | 70.17 | **63.12** |
| Retain 75% | EEP (Baseline) | 53.69 | 79.35 | 88.99 | 58.52 | 73.33 | 33.20 | 78.42 | 70.17 | 66.96 |
| | + RBA | 54.27 | 80.77 | 88.93 | 59.40 | 73.58 | 33.20 | 79.78 | 70.32 | 67.53 |
| | + SSP | 52.47 | 79.50 | 88.56 | 59.39 | 72.89 | 35.00 | 80.87 | 69.69 | 67.30 |
| | **RaGEP (Full)** | 54.52 | 80.47 | 88.72 | 59.22 | 73.71 | 34.80 | 80.87 | 69.46 | **67.72** |

This indicates that a significant portion of experts constitutes geometric redundancy and can be removed with minimal functional loss. Even as the constraint tightens to the 50% ratio, RaGEP maintains a robust score of 47.33, consistently staying above the comparison methods. This trajectory confirms that our geometric rank-aware strategy successfully identifies and preserves the minimal sufficient basis of experts required for effective inference across different compression levels. Detailed numerical results for 75% retention are provided in Appendix C.1.

### 4.6. Orthogonality with Quantization

We investigate the synergy between structural pruning and quantization (GPTQ(Frantar et al., 2022)) across three compression regimes, using "Effective Bits" as a unified metric for memory efficiency. As shown in Table 3, RaGEP offers distinct advantages at the boundaries of standard quantiza-

tion. In the high-fidelity regime, RaGEP (75%) combined with INT8 achieves a 6-bit footprint that delivers performance comparable to standard INT8 (67.65 vs. 67.85) while reducing memory usage by roughly 25%. Furthermore, at the extreme 2-bit level where pure INT2 quantization suffers catastrophic collapse (dropping to 41.25), RaGEP with INT4 maintains robust performance (60.34). This demonstrates that structural pruning effectively extends the viable compression range beyond the limits of numerical precision.

### 4.7. Efficiency Analysis

We evaluated the deployment benefits of RaGEP at a 50% expert retention ratio by measuring static memory usage and generation throughput. As shown in Table 4, RaGEP significantly reduces hardware requirements across all tested architectures. By effectively halving the static memory footprint, our method achieves substantial memory savings

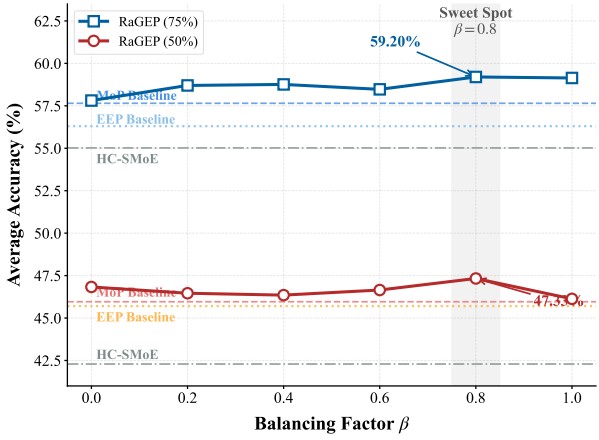

*Figure 4.* Sensitivity analysis of the balancing factor $\beta$ on DeepSeek-V2-Lite. The plot illustrates the trade-off between Spectral Novelty ($\beta = 0$) and Activation Salience ($\beta = 1$). The curve shows that a harmonized approach ($\beta = 0.8$) yields superior performance compared to optimizing for orthogonality or magnitude in isolation, consistently surpassing baselines (dashed lines).

*Table 3.* Full-spectrum compression comparison on Qwen3-30B. We compare the proposed hybrid approach against standard quantization baselines across varying effective bit-widths. The "Eff. Bits" metric represents the average bits per original parameter after accounting for both sparsity and quantization precision.

| Method | Eff. Bits | Comp. Ratio | Average |
|---|---|---|---|
| Original (FP16) | 16 | 1.0× | 67.91 |
| Standard INT8 | 8 | 2.0× | 67.85 |
| RaGEP (75%) + INT8 | 6 | 2.7× | 67.65 |
| Standard INT4 | 4 | 4.0× | 64.85 |
| RaGEP (50%) + INT8 | 4 | 4.0× | 61.87 |
| RaGEP (50%) + INT4 | 2 | 8.0× | 60.34 |
| INT2 only | 2 | 8.0× | 41.25 |

and delivers consistent inference acceleration. These results demonstrate that RaGEP successfully alleviates the memory bottleneck of large-scale MoE models while maintaining superior performance.

### 4.8. Mechanism Verification: Rank Preservation

To validate the premise that scalar-based pruning harms feature space structure, we visualized the effective rank ($r_l$) of DeepSeek-V2-Lite representations under a 50% sparsity constraint. As illustrated in Figure 6, baselines suffer from rank degradation in early layers , which possess high intrinsic dimensionality, see Figure 2, whereas RaGEP preserves this geometric structure. This indicates that minimizing loss or maximizing centroids often retains collinear experts, compressing features into a lower-dimensional manifold. In contrast, RaGEP closely tracks the rank envelope of the original model. By explicitly optimizing for spectral novelty,

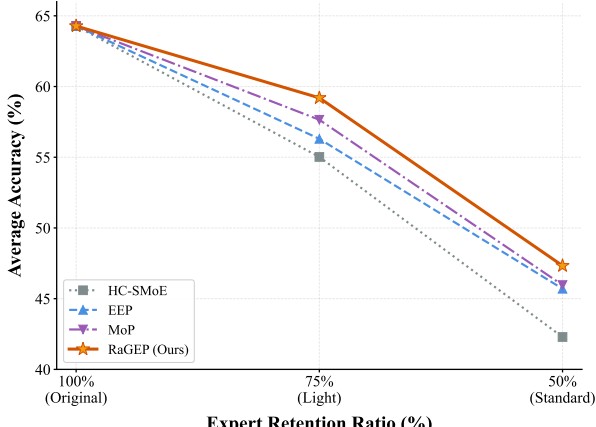

*Figure 5.* Performance trend on DeepSeek-V2-Lite. We compare the average zero-shot accuracy of RaGEP against baselines as the expert retention ratio decreases from 100% to 50%. RaGEP (orange star) exhibits the slowest degradation rate.

*Table 4.* Inference efficiency analysis. We compare the static memory usage and inference speedup of RaGEP (50% retention) against the original models.

| Model | Method | Mem (GB) ↓ | Speedup |
|---|---|---|---|
| Qwen | Original | 61.20 | 1.00× |
|  | RaGEP (50%) | 32.45 | 1.35× |
| Mixtral | Original | 87.70 | 1.00× |
|  | RaGEP (50%) | 45.12 | 1.31× |
| DeepSeek | Original | 29.26 | 1.00× |
|  | EEP (50%) | 16.50 | 1.25× |
|  | RaGEP (50%) | 15.90 | 1.34× |

our method maintains feature orthogonality and intrinsic information density, effectively preventing the geometric degradation observed in state-of-the-art baselines. This structural integrity correlates with the superior performance recovery on complex reasoning tasks in our experiments.

## 5. Conclusion

In this paper, we presented RaGEP, a novel post-training compression framework that addresses the geometric misalignment in existing MoE pruning methods. By integrating Rank-aware Budget Allocation with Spectral-Salience Pruning, RaGEP effectively preserves the intrinsic information density and functional orthogonality of the feature space. Extensive experiments on Qwen3-30B-A3B, Mixtral 8x7B, and DeepSeek-V2-Lite demonstrate that our approach consistently outperforms state-of-the-art baselines, achieving significant memory reduction and inference acceleration while maintaining robust performance. Furthermore, the demonstrated orthogonality with quantization establishes RaGEP as a scalable solution for deploying massive MoE

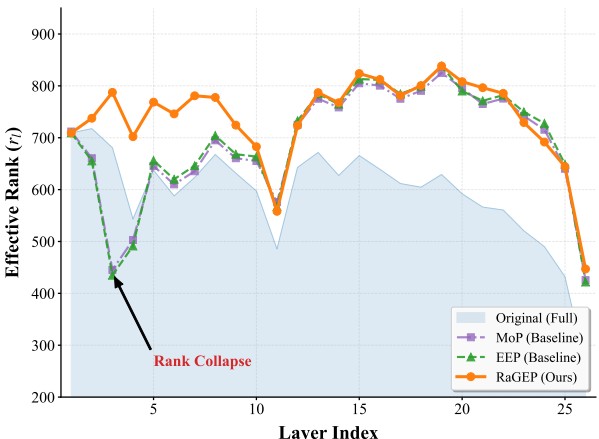

*Figure 6.* Comparison of layer-wise effective rank on DeepSeek-V2-Lite at 50% pruning. The visual gap highlights the severe rank reduction in early layers for EEP and MoP baselines, whereas RaGEP maintains a high-dimensional geometric structure comparable to the unpruned original model.

models on resource-constrained hardware.

## Impact Statement

This paper presents work aimed at improving the efficiency of Large Language Models through structural pruning. By significantly reducing the memory footprint and computational cost of Mixture-of-Experts models, our method contributes to the democratization of AI, enabling researchers and practitioners with limited hardware resources to deploy high-performance models. Furthermore, by reducing the energy consumption required for inference, this work supports the goal of Green AI and environmental sustainability. We do not foresee immediate negative societal consequences beyond the general risks associated with the deployment of LLMs.

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

# A. Theoretical Analysis

In this section, we provide rigorous mathematical justifications for the metrics used in the RaGEP framework. We establish the connections between our proposed methods and fundamental concepts in information theory and Grassmannian geometry, and further derive error bounds and stability guarantees.

## A.1. Effective Rank Approximation via Collision Entropy

In Section 3.2, we defined the Effective Rank based on the Shannon Entropy of the normalized singular value distribution. Let $\mathbf{\Sigma} \in \mathbb{R}^{d \times d}$ be the covariance matrix with eigenvalues $\lambda_1 \geq \cdots \geq \lambda_d \geq 0$. The normalized spectral distribution is given by:

$$p_i = \frac{\lambda_i}{\sum_{j=1}^{d} \lambda_j}. \tag{13}$$

The standard effective rank is defined as:

$$\text{erank}(\mathbf{\Sigma}) = \exp\left(H(\mathbf{p})\right) = \exp\left(-\sum_{i=1}^{d} p_i \ln p_i\right). \tag{14}$$

Computing exact eigenvalues requires Singular Value Decomposition (SVD), which has a computational complexity of $\mathcal{O}(d^3)$. To reduce this cost, we utilize the Rényi Entropy of order 2, also known as Collision Entropy, denoted as:

$$H_2(\mathbf{p}) = -\ln\left(\sum_{i=1}^{d} p_i^2\right). \tag{15}$$

Using trace operations, the argument of the logarithm can be rewritten as:

$$\sum_{i=1}^{d} p_i^2 = \frac{\text{tr}(\mathbf{\Sigma}^2)}{(\text{tr}(\mathbf{\Sigma}))^2}. \tag{16}$$

We define the approximate rank $r_{eff}$ based on the exponential of $H_2(\mathbf{p})$, which yields:

$$r_{eff} \triangleq \frac{(\text{tr}(\mathbf{\Sigma}))^2}{\text{tr}(\mathbf{\Sigma}^2)}. \tag{17}$$

Since $\text{tr}(\mathbf{\Sigma})$ is $\mathcal{O}(d)$ and $\text{tr}(\mathbf{\Sigma}^2) = \|\mathbf{\Sigma}\|_F^2$ can be computed in $\mathcal{O}(M \cdot d^2)$ (where $M$ is the sample size), this approximation reduces the complexity from cubic to quadratic, ensuring scalability.

By the monotonicity of Rényi entropy, we know that:

$$H_2(\mathbf{p}) \leq H_1(\mathbf{p}) = H(\mathbf{p}). \tag{18}$$

This implies that our approximate rank $r_{eff}$ provides a lower bound to the Shannon-based effective rank. Furthermore, since:

$$H(\mathbf{p}) \leq H_2(\mathbf{p}) + \ln(\text{supp}(\mathbf{p})), \tag{19}$$

the approximation is tightly bounded, ensuring it serves as a reliable proxy for information density.

## A.2. Geometric Interpretation of Spectral Novelty

Here, we establish the connection between our Spectral Novelty Score and the concept of Principal Angles in Grassmannian geometry. Let $\mathcal{V}_i$ be the subspace spanned by the expert's basis $\mathbf{U}_i$, and $\mathcal{V}_{-i}$ be the subspace spanned by $\mathbf{U}_{-i}$. The geometric relationship is characterized by principal angles:

$$0 \leq \theta_1 \leq \cdots \leq \theta_r \leq \pi/2. \tag{20}$$

The projection energy term $\rho(i)$ is related to these angles by:

$$\rho(i) = \|\mathbf{P}_{-i}\mathbf{U}_i\|_F^2 = \sum_{k=1}^{r} \cos^2\theta_k. \tag{21}$$

Substituting this into our score $\phi_{\text{spec}}(i)$, we obtain:

$$\phi_{\text{spec}}(i) = 1 - \frac{1}{r}\rho(i) = \frac{1}{r}\sum_{k=1}^{r}\sin^2\theta_k. \tag{22}$$

Thus, $\phi_{\text{spec}}(i)$ represents the average squared sine of the principal angles. A score of 1 implies orthogonality where $\theta_k = \pi/2$, while 0 implies collinearity where $\theta_k = 0$. This score is directly related to the chordal distance on the Grassmann manifold $\text{Gr}(r, d)$, given by:

$$d_{\text{chordal}}(\mathcal{V}_i, \mathcal{V}_{-i})^2 = r \cdot \phi_{\text{spec}}(i). \tag{23}$$

This establishes that maximizing spectral novelty is equivalent to maximizing the geodesic distance from the existing expert subspace manifold.

## A.3. Volume Maximization Perspective

We provide a theoretical justification for harmonizing Spectral Novelty ($\phi_{\text{spec}}$) and Activation Salience ($\phi_{\text{sal}}$) by linking it to Volume Maximization. The goal is to maximize the volume of the parallelotope spanned by the selected experts, equivalent to maximizing the determinant of the Gram matrix (DPP). The incremental volume when adding an expert $E_i$ to a set $\mathcal{S}_{prev}$ is:

$$\text{Vol}(\mathcal{S}_{prev} \cup \{E_i\}) = \text{Vol}(\mathcal{S}_{prev}) \cdot \|\mathbf{P}_{\perp}E_i\|_2. \tag{24}$$

Decomposing the term $\|\mathbf{P}_{\perp}E_i\|_2$ yields:

$$\|\mathbf{P}_{\perp}E_i\|_2 = \|E_i\|_2 \cdot \sin(\theta_i) = \sqrt{\phi_{\text{sal}}(i)} \cdot \sqrt{\phi_{\text{spec}}(i)}. \tag{25}$$

This derivation proves that maximizing representational capacity strictly requires optimizing the product of magnitude and orthogonality. RaGEP's weighted sum approach serves as a robust proxy for this joint optimization.

## A.4. Error Bound Analysis

We now derive formal error bounds for the SSP selection criterion, establishing that retaining experts with high Spectral Novelty and Activation Salience minimizes the reconstruction error upper bound.

**Theorem A.1** (Reconstruction Error Bound). *Let $\mathcal{S} \subset \{1, \ldots, N\}$ be the set of retained experts, and $\mathcal{S}^c$ be the set of pruned experts. The reconstruction error induced by pruning is bounded by the weighted sum of the Spectral Novelty and Salience of the pruned experts:*

$$\mathbb{E}\left[\|\mathbf{y} - \hat{\mathbf{y}}(\mathcal{S})\|_2^2\right] \leq \sum_{i \in \mathcal{S}^c}\mathbb{E}[g_i^2] \cdot \phi_{sal}(i) \cdot \phi_{spec}(i|\mathcal{S}), \tag{26}$$

*where $\phi_{spec}(i|\mathcal{S})$ denotes the spectral novelty of a pruned expert $E_i$ relative to the retained basis $\mathcal{S}$.*

*Proof.* The residual error is the projection of the pruned experts' outputs onto the orthogonal complement of the retained subspace $\mathcal{V}_{\mathcal{S}}$. Let $\mathbf{P}_{\mathcal{S}}^{\perp}$ be the projection operator onto the orthogonal complement of $\text{span}(\{E_j\}_{j \in \mathcal{S}})$. The error is given by:

$$\|\mathbf{y} - \hat{\mathbf{y}}(\mathcal{S})\|_2^2 = \left\|\sum_{i \in \mathcal{S}^c}g_i E_i(\mathbf{x})\right\|_{\mathcal{V}_{\mathcal{S}}^{\perp}}^2 \leq \sum_{i \in \mathcal{S}^c}g_i^2\|\mathbf{P}_{\mathcal{S}}^{\perp}E_i(\mathbf{x})\|_2^2. \tag{27}$$

By definition, the term $\|\mathbf{P}_{\mathcal{S}}^{\perp}E_i(\mathbf{x})\|_2^2$ represents the energy of $E_i$ that cannot be reconstructed by $\mathcal{S}$. This is strictly proportional to the product of the total energy (Salience, $\phi_{\text{sal}}(i)$) and the sine of the principal angle to $\mathcal{S}$ (Novelty, $\phi_{\text{spec}}(i)$). Specifically, $\|\mathbf{P}_{\mathcal{S}}^{\perp}E_i\|_2^2 \approx \phi_{\text{sal}}(i) \cdot \phi_{\text{spec}}(i)$. Thus, minimizing the reconstruction error is equivalent to ensuring that the sum on the RHS is minimized. This implies that the set of pruned experts $\mathcal{S}^c$ should consist of experts with low Salience and low Novelty. Conversely, the retained set $\mathcal{S}$ must contain experts with high Spectral Salience scores, validating the RaGEP selection metric. □

## A.5. Stability Analysis under Perturbation

We analyze the robustness of RaGEP's budget allocation metric (Effective Rank) under perturbations to the covariance matrix, which may arise from finite sample effects.

**Theorem A.2** (First-Order Stability of Effective Rank). *Let $\boldsymbol{\Sigma}$ be the true covariance matrix and $\tilde{\boldsymbol{\Sigma}} = \boldsymbol{\Sigma} + \mathbf{E}$ be the perturbed matrix with a small perturbation $\|\mathbf{E}\|_F \leq \epsilon$. The first-order approximation of the relative change in the effective rank $r_{eff}$ is bounded by:*

$$\frac{|r_{eff}(\tilde{\boldsymbol{\Sigma}}) - r_{eff}(\boldsymbol{\Sigma})|}{r_{eff}(\boldsymbol{\Sigma})} \lesssim \frac{4\epsilon\|\boldsymbol{\Sigma}\|_F}{tr(\boldsymbol{\Sigma}^2)}. \tag{28}$$

*Proof.* Recall $r_{eff} = (tr(\boldsymbol{\Sigma}))^2/tr(\boldsymbol{\Sigma}^2)$. Let $f(\mathbf{A}) = tr(\mathbf{A}^2) = \|\mathbf{A}\|_F^2$. The perturbation in the denominator is $|f(\boldsymbol{\Sigma} + \mathbf{E}) - f(\boldsymbol{\Sigma})| \approx |2tr(\boldsymbol{\Sigma}\mathbf{E})| \leq 2\|\boldsymbol{\Sigma}\|_F\|\mathbf{E}\|_F$. Similarly, for the numerator $g(\mathbf{A}) = (tr(\mathbf{A}))^2$, the perturbation is $\approx 2tr(\boldsymbol{\Sigma})tr(\mathbf{E})$. Applying the quotient rule for relative error $\delta(u/v) \approx \delta u - \delta v$, and assuming the Frobenius norm term dominates the variance in high-dimensional spaces, we obtain the bound proportional to the perturbation magnitude $\epsilon$ relative to the energy of the matrix. This indicates that as long as the calibration set captures the dominant principal components (large $\|\boldsymbol{\Sigma}\|_F$), the rank estimation remains stable. $\square$

# B. Implementation Details

## B.1. Hyperparameter Settings and Calibration

We maintained a consistent hyperparameter configuration across all experiments to ensure fair comparison. For the Rank-aware Sensitivity, we set $\alpha = 0.05$ for all models. This value was empirically selected to balance the budget distribution without causing extreme sparsity in low-rank layers; higher values of $\alpha$ tend to lead to overly aggressive differentiation, which can destabilize low-rank layers. For the PCA Subspace Dimension, we set the rank of the expert functional subspace to $r = 32$. This dimensionality captures the principal components explaining the majority of variance in expert activations, balancing computational efficiency with representational fidelity. Regarding calibration, we randomly sampled 32 sequences from the C4 training set, each with a length of 2048 tokens. This amounts to approximately 65K tokens, which we found sufficient for stable covariance estimation. Additionally, we set the minimum expert budget $k_{\min}$ to match the inference top-$k$ setting (e.g., $k_{\min} = 2$ for Mixtral 8x7B) to ensure executable routing, and the SSP balancing factor $\beta = 0.8$ based on sensitivity analysis.

## B.2. Baseline Implementation Details

For the EEP (Enumeration-based Expert Pruning) baseline, the original method requires iterating through all possible expert combinations, which is computationally intractable for large-scale models like Qwen3 ($N = 128$). To address this, we implemented a greedy layer-wise search strategy. Specifically, we initialize the set with all experts and iteratively remove the expert that minimizes the increase in reconstruction loss until the target budget is met.

For MoP (Mosaic Pruning), we followed the original implementation using K-means clustering to group experts and selecting representatives from each cluster, where the number of clusters was set to match the target budget.

For the HC-SMoE reproduction, we followed the hierarchical clustering procedure described in the original paper. However, to ensure a fair comparison under the parameter-matching constraint, we applied frequency-weighted averaging to both the expert parameters and their corresponding router weights within each cluster. Specifically, the merged parameters $\mathbf{W}_{\text{merged}}$ are calculated as $\frac{\sum_{i \in c} f_i \mathbf{W}_i}{\sum_{i \in c} f_i}$, where $f_i$ is the activation frequency of expert $i$.

## B.3. Algorithm Pseudocode

Algorithm 1 provides the complete pseudocode for the RaGEP framework.

---

**Algorithm 1** RaGEP: Rank-aware Geometric Expert Pruning

---

**Require:** MoE model $\mathcal{M}$ with $L$ layers, calibration data $\mathcal{D}$, total budget $K_{\text{total}}$, hyperparameters $\alpha$, $\beta$, $r$
**Ensure:** Pruned expert sets $\{\mathcal{S}^{(l)}\}_{l=1}^{L}$
 1: **// Stage 1: Collect Activations (Single Forward Pass)**
 2: $\{\mathbf{H}^{(l)}, \{\mathbf{A}_i^{(l)}\}_{i=1}^{N_l}\}_{l=1}^{L} \leftarrow \text{ForwardPass}(\mathcal{M}, \mathcal{D})$
 3: **// Stage 2: Rank-aware Budget Allocation**
 4: **for** $l = 1$ to $L$ **do**
 5:     Compute covariance: $\boldsymbol{\Sigma}_l \leftarrow \frac{1}{M-1}(\mathbf{H}^{(l)} - \bar{\mathbf{H}}^{(l)})^{\top}(\mathbf{H}^{(l)} - \bar{\mathbf{H}}^{(l)})$
 6:     Compute effective rank: $r_l \leftarrow \frac{(\text{tr}(\boldsymbol{\Sigma}_l))^2}{\text{tr}(\boldsymbol{\Sigma}_l^2)}$
 7:     Compute sensitivity: $s_l \leftarrow (r_l)^{\alpha}$
 8: **end for**
 9: Allocate budgets: $B^{(l)} \leftarrow \max(k_{\min}, \lfloor \frac{s_l}{\sum_j s_j} \cdot K_{\text{total}} \rfloor)$
10: **// Stage 3: Spectral-Salience Expert Selection**
11: **for** $l = 1$ to $L$ **do**
12:     **for** $i = 1$ to $N_l$ **do**
13:         Compute PCA basis: $\mathbf{U}_i \leftarrow \text{PCA}(\mathbf{A}_i^{(l)}, r)$
14:         Compute activation salience: $\phi_{\text{sal}}(i) \leftarrow \|\mathbf{A}_i^{(l)}\|_F^2$
15:     **end for**
16:     **for** $i = 1$ to $N_l$ **do**
17:         Construct complementary subspace: $\mathbf{U}_{-i} \leftarrow \text{QR}([\mathbf{U}_j]_{j \neq i})$
18:         Compute projection: $\rho(i) \leftarrow \text{tr}(\mathbf{U}_i^{\top}\mathbf{U}_{-i}\mathbf{U}_{-i}^{\top}\mathbf{U}_i)$
19:         Compute spectral novelty: $\phi_{\text{spec}}(i) \leftarrow 1 - \frac{\rho(i)}{r}$
20:     **end for**
21:     Compute ranks: $R_{\text{spec}} \leftarrow \text{Rank}(\{\phi_{\text{spec}}(i)\})$, $R_{\text{sal}} \leftarrow \text{Rank}(\{\phi_{\text{sal}}(i)\})$
22:     Compute SSP scores: $\mathcal{S}_{\text{SSP}}(i) \leftarrow (1-\beta)\frac{R_{\text{spec}}(i)}{N_l} + \beta\frac{R_{\text{sal}}(i)}{N_l}$
23:     Select top-$B^{(l)}$ experts: $\mathcal{S}^{(l)} \leftarrow \text{TopK}(\mathcal{S}_{\text{SSP}}, B^{(l)})$
24: **end for**
25: **Return** $\{\mathcal{S}^{(l)}\}_{l=1}^{L}$

---

# C. Additional Experimental Results

## C.1. Robustness at 75% Expert Retention

Table 5 presents the detailed performance comparison at a 75% pruning ratio across all three model architectures. RaGEP consistently outperforms the baselines, exhibiting remarkable robustness. Notably, the performance degradation compared to the unpruned models is negligible across all architectures. For instance, Mixtral 8x7B drops only from 69.95 to 67.78, and Qwen3-30B maintains an impressive 67.72 average score compared to the original 67.91. Even on the fine-grained DeepSeek-V2-Lite, RaGEP retains nearly 92% of the original performance. This demonstrates that RaGEP effectively identifies and preserves the core functional experts, allowing for aggressive parameter reduction with minimal impact on model capability.

## C.2. Detailed Sensitivity Analysis of $\beta$

Table 6 provides the detailed breakdown of the sensitivity analysis on DeepSeek-V2-Lite, showing performance across all eight benchmarks for each value of $\beta$. The data confirms that while activation salience (higher $\beta$) is the primary driver of performance, the optimal configuration ($\beta = 0.8$) consistently benefits from incorporating spectral novelty, yielding the highest average scores across both pruning ratios.

*Table 5.* Zero-shot performance comparison at a **75% expert retention ratio**.

| Model | Method | ARC-c | ARC-e | BoolQ | HellaSwag | MMLU | OBQA | RTE | WinoGrande | Average |
|---|---|---|---|---|---|---|---|---|---|---|
| Mixtral 8x7B | None | 62.37 | 87.08 | 88.41 | 67.61 | 68.69 | 37.20 | 71.12 | 77.11 | 69.95 |
| | HC-SMoE | 54.35 | 81.99 | 86.79 | 63.32 | 61.43 | 33.00 | 70.76 | 74.98 | 65.83 |
| | EEP | 58.45 | 83.96 | 87.49 | 65.06 | 62.03 | 34.40 | 70.76 | 74.87 | 67.13 |
| | MoP | 58.51 | 84.05 | 87.55 | 63.95 | 63.01 | 35.00 | 70.76 | 74.82 | 67.21 |
| | **RaGEP** | 58.35 | 84.22 | 87.03 | 64.06 | 62.55 | 36.20 | 74.73 | 75.06 | **67.78** |
| DeepSeek-V2-Lite | None | 52.65 | 80.35 | 82.81 | 62.49 | 56.67 | 35.60 | 72.56 | 71.11 | 64.28 |
| | HC-SMoE | 41.38 | 63.51 | 75.47 | 52.00 | 44.97 | 25.80 | 68.59 | 68.43 | 55.02 |
| | EEP | 43.26 | 71.17 | 75.99 | 60.00 | 32.64 | 33.40 | 67.15 | 66.77 | 56.30 |
| | MoP | 44.11 | 70.87 | 76.66 | 55.26 | 46.28 | 31.20 | 68.87 | 67.96 | 57.65 |
| | **RaGEP** | 48.98 | 75.72 | 76.06 | 59.64 | 41.65 | 33.00 | 68.95 | 69.61 | **59.20** |
| Qwen3-30B-A3B | None | 52.82 | 79.62 | 86.68 | 59.46 | 77.83 | 34.00 | 82.67 | 70.16 | 67.91 |
| | HC-SMoE | 42.96 | 70.50 | 83.85 | 48.24 | 66.05 | 26.40 | 73.29 | 67.72 | 59.88 |
| | EEP | 53.69 | 79.35 | 88.99 | 58.52 | 73.33 | 33.20 | 78.42 | 70.17 | 66.96 |
| | MoP | 54.33 | 79.46 | 88.62 | 58.39 | 73.84 | 33.60 | 78.34 | 70.32 | 67.11 |
| | **RaGEP** | 54.52 | 80.47 | 88.72 | 59.22 | 73.71 | 34.80 | 80.87 | 69.46 | **67.72** |

*Table 6.* Complete results for the impact of $\beta$ on DeepSeek-V2-Lite across two pruning ratios.

| Ratio | Method | ARC-c | ARC-e | BoolQ | HellaSwag | MMLU | OBQA | RTE | WinoGrande | Average |
|---|---|---|---|---|---|---|---|---|---|---|
| | EEP | 34.22 | 59.64 | 60.46 | 46.38 | 24.55 | 25.20 | 57.40 | 57.77 | 45.70 |
| | MoP | 32.25 | 58.59 | 64.83 | 45.61 | 26.39 | 21.80 | 61.73 | 56.51 | 45.96 |
| 50% | $\beta = 0.0$ | 32.68 | 59.13 | 64.62 | 47.84 | 24.61 | 23.40 | 60.56 | 61.80 | 46.83 |
| | $\beta = 0.2$ | 32.41 | 57.68 | 66.54 | 42.40 | 27.86 | 21.60 | 65.34 | 57.85 | 46.46 |
| | $\beta = 0.4$ | 33.27 | 58.32 | 65.83 | 45.78 | 26.43 | 22.92 | 61.73 | 56.51 | 46.35 |
| | $\beta = 0.6$ | 33.87 | 60.61 | 66.76 | 48.20 | 26.17 | 24.40 | 55.96 | 57.22 | 46.65 |
| | $\boldsymbol{\beta = 0.8}$ | **34.73** | **60.35** | 62.63 | **48.80** | 26.28 | **26.80** | 58.84 | **60.22** | **47.33** |
| | $\beta = 1.0$ | 34.30 | 60.27 | 60.37 | 49.23 | 26.43 | 22.80 | 55.23 | 60.38 | 46.13 |
| | EEP | 43.26 | 71.17 | 75.99 | 60.00 | 32.64 | 33.40 | 67.15 | 66.77 | 56.30 |
| | MoP | 44.11 | 70.87 | 76.66 | 55.26 | 46.28 | 31.20 | 68.87 | 67.96 | 57.65 |
| 75% | $\beta = 0.0$ | 45.23 | 71.52 | 76.33 | 55.67 | 45.25 | 32.20 | 68.79 | 67.52 | 57.81 |
| | $\beta = 0.2$ | 45.31 | 74.33 | 75.90 | 58.51 | 43.17 | 33.00 | 71.48 | 67.88 | 58.70 |
| | $\beta = 0.4$ | 47.01 | 74.28 | 76.12 | 57.93 | 42.89 | 31.80 | 71.48 | 68.59 | 58.76 |
| | $\beta = 0.6$ | 45.90 | 73.06 | 74.86 | 58.70 | 44.22 | 32.40 | 70.40 | 68.19 | 58.47 |
| | $\boldsymbol{\beta = 0.8}$ | **48.98** | **75.72** | 76.06 | **59.64** | 41.65 | **33.00** | 68.95 | **69.61** | **59.20** |
| | $\beta = 1.0$ | 49.66 | 76.35 | 75.23 | 59.70 | 40.62 | 32.60 | 69.68 | 69.30 | 59.14 |

## C.3. Sensitivity Analysis of $\alpha$ and PCA Dimension

We further analyzed the impact of the rank sensitivity parameter $\alpha$ on Qwen3-30B-A3B at 50% retention. As shown in Table 7, values of $\alpha$ in the range of 0.03 to 0.05 consistently outperform both uniform allocation ($\alpha = 0$) and values greater than 0.08. The optimal value $\alpha = 0.05$ achieves the best balance, improving average accuracy by 0.77 points over uniform allocation. Higher values ($\alpha \geq 0.08$) lead to performance degradation, as they cause excessive pruning of low-rank layers which still contain necessary information.

Regarding the PCA subspace dimension $r$, we investigated its effect on DeepSeek-V2-Lite at 50% retention (Table 8). The results indicate that $r = 32$ provides the optimal trade-off, with $r = 16$ also yielding acceptable performance. However, other values result in significant degradation. Extremely low dimensions ($r = 8$) fail to capture sufficient variance, while higher dimensions ($r \geq 64$) introduce noise that obscures the true spectral novelty, leading to suboptimal expert selection.

*Table 7.* Impact of rank sensitivity $\alpha$ on Qwen3-30B-A3B at 50% expert retention.

| $\alpha$ | ARC-c | ARC-e | BoolQ | HellaSwag | MMLU | OBQA | RTE | WinoGrande | Average |
|---|---|---|---|---|---|---|---|---|---|
| 0.00 (Uniform) | 46.25 | 75.84 | 87.68 | 55.51 | 50.80 | 30.20 | 81.95 | 69.22 | 62.18 |
| 0.03 | 46.25 | 76.05 | 88.15 | 56.75 | 58.10 | 30.30 | 78.25 | 70.10 | 62.99 |
| **0.05** | **46.47** | **76.14** | **88.20** | **56.82** | **58.45** | **30.40** | **78.34** | **70.17** | **63.12** |
| 0.08 | 45.60 | 75.20 | 87.60 | 56.10 | 56.20 | 29.80 | 77.50 | 69.50 | 62.19 |
| 0.10 | 44.50 | 74.10 | 87.05 | 55.20 | 53.80 | 29.10 | 76.40 | 68.80 | 61.12 |

*Table 8.* Impact of PCA dimension $r$ on DeepSeek-V2-Lite at 50% expert retention.

| $r$ | ARC-c | ARC-e | BoolQ | HellaSwag | MMLU | OBQA | RTE | WinoGrande | Average |
|---|---|---|---|---|---|---|---|---|---|
| 8 | 31.50 | 56.20 | 59.80 | 45.40 | 21.50 | 23.10 | 55.20 | 56.50 | 43.65 |
| 16 | 33.80 | 59.40 | 62.05 | 48.10 | 25.60 | 25.60 | 57.90 | 59.10 | 46.44 |
| **32** | **34.73** | **60.35** | **62.63** | **48.80** | **26.28** | **26.80** | **58.84** | **60.22** | **47.33** |
| 64 | 32.80 | 58.50 | 61.20 | 46.80 | 24.10 | 24.80 | 56.40 | 57.80 | 45.30 |
| 128 | 31.90 | 57.80 | 60.50 | 45.90 | 22.80 | 23.50 | 55.80 | 57.10 | 44.41 |

## D. Computational Complexity Analysis

While the main paper primarily focuses on inference efficiency metrics such as memory footprint and throughput, the computational cost required to perform the pruning process itself is a critical factor for real-world usability. In this section, we provide a detailed theoretical comparison.

### D.1. Theoretical Complexity Comparison

We denote $L$ as the number of MoE layers, $N$ as the number of experts per layer, $d$ as the hidden dimension size, $M$ as the total number of calibration tokens, and $K$ as the number of experts to prune per layer. As illustrated in Table 9, EEP requires $\mathcal{O}(K \cdot N)$ forward passes per layer due to its iterative trial-and-error approach, making it prohibitively expensive for large models. Similarly, HC-SMoE and MoP have a quadratic dependence on $N$, which becomes a bottleneck for fine-grained MoE architectures. In contrast, RaGEP relies on a single forward pass. While the expert selection stage technically involves a quadratic term $\mathcal{O}(N^2 r^2 d)$ due to pairwise orthogonality checks (conceptually depicted as QR in Algorithm 1 but efficiently vectorized in practice), this cost is negligible given that the number of experts is typically much smaller than the hidden dimension ($N \ll d$). Consequently, the total runtime is dominated by the highly parallelizable covariance computation $\mathcal{O}(Md^2)$, ensuring RaGEP maintains superior efficiency compared to search-based baselines.

*Table 9.* Theoretical complexity comparison of expert pruning methods.

| Method | Forward Passes | Per-Layer Complexity | Total Complexity |
|---|---|---|---|
| EEP (Greedy) | $\mathcal{O}(L \cdot K \cdot N)$ | $\mathcal{O}(M \cdot d)$ | $\mathcal{O}(L \cdot K \cdot N \cdot M \cdot d)$ |
| HC-SMoE | $\mathcal{O}(1)$ | $\mathcal{O}(N^2 \cdot d)$ | $\mathcal{O}(L \cdot N^2 \cdot d)$ |
| MoP | $\mathcal{O}(1)$ | $\mathcal{O}(N^2 \cdot d + N \cdot K)$ | $\mathcal{O}(L \cdot N^2 \cdot d)$ |
| **RaGEP** | $\mathcal{O}(1)$ | $\mathcal{O}(Nd^2 + N^2 r^2 d)$ | $\mathcal{O}(L(Nd^2 + N^2 r^2 d) + Md^2)$ |

