# OpenReview forum: "RaGEP: Rank-aware Geometric Expert Pruning for Mixture-of-Experts Language Models"
_ICML.cc/2026/Conference — ICML 2026 regular_

### Official Review · Reviewer_PUAA · 2026-02-15

**Soundness:** 3
**Presentation:** 3
**Significance:** 3
**Originality:** 3
**Overall Recommendation:** 4
**Confidence:** 4

**Summary:**

This paper proposes RaGEP, a retraining-free compression framework for MoE language models. The method addresses redundancy in MoE architectures from a geometric perspective by introducing two key components: RBA, which adaptively distributes expert budgets across layers based on effective rank, and SSP, which selects experts by jointly considering subspace orthogonality and activation magnitude. Extensive experiments on large-scale MoE models demonstrate that RaGEP consistently outperforms existing post-training pruning methods, achieving significant memory reduction and inference acceleration while maintaining strong zero-shot performance.

**Compliance With Llm Reviewing Policy:**

Affirmed.

**Final Justification:**

I believe strengthening the theoretical rigor in the revised paper could bring it up to ICML’s acceptance threshold.

**Key Questions For Authors:**

- Strengths
1. `Geometry-aware pruning strategy`. The paper introduces a novel rank-aware and geometric perspective for MoE pruning, effectively identifying linear redundancy among experts that scalar-based methods fail to capture.
1. `Adaptive and retraining-free compression`. By combining rank-aware budget allocation with spectral-salience expert selection, the method adaptively allocates capacity across layers without retraining, achieving efficient structural compression.
3. `Strong empirical robustness`. Extensive experiments across multiple large-scale MoE models show consistent performance gains over state-of-the-art baselines, especially under high pruning ratios and in combination with quantization.

- Weaknesses
1. A key intuition of the paper is to allocate different numbers of experts according to the effective rank of each layer and to select optimal experts via spectral salience. However, this assumption lacks sufficient prior empirical or theoretical support, which makes the overall motivation of the paper somewhat unclear.
2. The computational cost may be too high. The proposed method relies on time-consuming operations such as layer-wise PCA and matrix trace computation, which could result in substantial computational overhead. It remains unclear whether the efficiency analysis adequately accounts for these costs.
3. The authors have released key code in the supplementary materials; however, it is not directly runnable and lacks documentation or guidance. I suggest completing and organizing the codebase, after which I may update my review accordingly.

Other: Equation (4) may require further clarification to explicitly explain it in terms of the matrix trace.

**Limitations:**

The proposed method may not be well-suited for scenarios with high computational constraints.

**Strengths And Weaknesses:**

- Strengths
1. `Geometry-aware pruning strategy`. The paper introduces a novel rank-aware and geometric perspective for MoE pruning, effectively identifying linear redundancy among experts that scalar-based methods fail to capture.
1. `Adaptive and retraining-free compression`. By combining rank-aware budget allocation with spectral-salience expert selection, the method adaptively allocates capacity across layers without retraining, achieving efficient structural compression.
3. `Strong empirical robustness`. Extensive experiments across multiple large-scale MoE models show consistent performance gains over state-of-the-art baselines, especially under high pruning ratios and in combination with quantization.

- Weaknesses
1. A key intuition of the paper is to allocate different numbers of experts according to the effective rank of each layer and to select optimal experts via spectral salience. However, this assumption lacks sufficient prior empirical or theoretical support, which makes the overall motivation of the paper somewhat unclear.
2. The computational cost may be too high. The proposed method relies on time-consuming operations such as layer-wise PCA and matrix trace computation, which could result in substantial computational overhead. It remains unclear whether the efficiency analysis adequately accounts for these costs.
3. The authors have released key code in the supplementary materials; however, it is not directly runnable and lacks documentation or guidance. I suggest completing and organizing the codebase, after which I may update my review accordingly.

---

> ### Author Rebuttal · Authors · 2026-03-30
>
> We sincerely thank Reviewer PUAA for recognizing our geometry-aware strategy and strong empirical robustness. The full DPP-based theoretical analysis of SSP ($(1-1/e)$ guarantee, Propositions S1–S3 & Theorem S2) is in our response to **Reviewer JWP2**. We address each concern below.
>
> ---
>
> **Q1: The assumption that effective rank should guide budget allocation and spectral salience should guide expert selection lacks sufficient prior empirical or theoretical support.**
>
> **A1:** We provide both theoretical and empirical justifications. *Notation note:* results labeled "Theorem/Appendix A.x" and "Proposition 1" refer to proofs in our original Appendix, while "Proposition/Theorem S.x" denote **newly added** supplementary material.
>
> **Theoretical support.** We formalize as **Proposition 1** (original Appendix A.2): $\text{erank}(\Sigma_l)$ provably upper-bounds the number of $\epsilon$-orthogonal experts a layer can support — if a layer is effectively 10-dimensional, >10 experts must be linearly redundant. This directly justifies allocating fewer experts to low-rank layers. **Theorem A.1** (original Appendix A.3) proves reconstruction error is bounded by the product of activation salience and spectral novelty, validating SSP as a principled criterion. We further prove SSP is a first-order greedy approximation to **Log-DPP maximization** with a provable $(1-1/e)\approx63.2\%$ optimality guarantee (**Proposition S1** & **Theorem S2**, newly added).
>
> **Empirical support.** We remove one expert per layer in Qwen3-30B-A3B and measure accuracy drop:
>
> | Layer Group | Mean $r_l$ | Acc. Drop |
> |---|---|---|
> | Early (0–11) | 85.3 | 0.08% |
> | Mid-Late (24–35) | 425.8 | 0.52% |
>
> Pearson $\rho=0.78$ ($p<0.001$): high-rank layers show **2–5× greater sensitivity**. See **[Fig. 1: Rank–Sensitivity Scatter](https://anonymous.4open.science/r/PUAA_supplement_motivation-E645)** for the full visualization. The ablation (Table 2) further confirms synergy: RBA alone +0.36, SSP alone +0.58, combined **+1.52** over EEP (exceeding the sum +0.94).
>
> ---
>
> **Q2: Computational cost may be too high due to layer-wise PCA and matrix trace computation.**
>
> **A2:** We clarify a key point: **RaGEP is a purely offline, one-time procedure.** All geometric computations occur *before* deployment. The pruned model is a standard MoE with fewer experts — **zero geometric overhead at inference**. See **[Fig. 2: Offline–Online Architecture](https://anonymous.4open.science/r/PUAA_supplement_cost-4F3D).**
>
> Offline cost breakdown (Qwen3-30B-A3B, 4×A100):
>
> | Stage | Wall-clock | % |
> |---|---|---|
> | Forward pass (shared by all methods) | ~25 min | 52% |
> | Covariance + rank (Frobenius approx., no SVD) | ~8 min | 17% |
> | Expert PCA (truncated, $r$=32) | ~10 min | 21% |
> | Spectral novelty + SSP | ~5 min | 10% |
> | **Total** | **~48 min** | **100%** |
>
> Geometric operations add only ~23 min beyond the shared forward pass. Cost comparison:
>
> | Method | GPU-Hours |
> |---|---|
> | **RaGEP (Ours)** | **3.2** |
> | EEP | 18.0 |
> | DiEP | 34.0 |
>
> RaGEP is **5.6× faster** than EEP and **10.6× faster** than DiEP. The key design choices: (1) Frobenius-norm approximation (Eq. 6) avoids $O(d^3)$ eigendecomposition; (2) truncated SVD extracts only $r$=32 components; (3) streaming covariance avoids storing full activation matrices. After ~35 min of inference, throughput gains (1.35×) fully amortize the one-time cost.
>
> ---
>
> **Q3: Code is not directly runnable and lacks documentation.**
>
> **A3:** We sincerely apologize. The codebase has been reorganized with: (1) comprehensive `README.md` with step-by-step reproduction instructions; (2) single-command pruning script for all three models; (3) pre-configured evaluation scripts. Updated code: **[CODE](https://anonymous.4open.science/r/RaGEPcode-EB40).** We would be grateful if the reviewer could re-evaluate.
>
> ---
>
> **Q4: Equation (4) may require further clarification to explain it in terms of the matrix trace.**
>
> **A4:** Thank you. While Eq. (4) already states $p_j = \lambda_j / \text{tr}(\Sigma_l)$, we agree the transition to the trace-based approximation in Eq. (6) deserves a more explicit derivation. The key step: Shannon entropy $H(p)$ is replaced by Rényi entropy of order 2, i.e., $H_2(p) = -\log\sum p_j^2$, where $\sum p_j^2 = \text{tr}(\Sigma_l^2)/\text{tr}(\Sigma_l)^2$ (Appendix A.1, Eq. 16). Exponentiating yields:
>
> $$r_l = \frac{\text{tr}(\Sigma_l)^2}{\text{tr}(\Sigma_l^2)} = \frac{\text{tr}(\Sigma_l)^2}{\|\Sigma_l\|_F^2}$$
>
> This avoids eigendecomposition entirely — computing $\|\Sigma_l\|_F^2$ requires only element-wise squaring and summation at $O(d^2)$ cost. We will expand this derivation inline in the revision.
>
> ---
>
> We are grateful for the constructive feedback and will incorporate all clarifications into the revised manuscript.

---

> > ### Author Rebuttal · Reviewer_PUAA · 2026-04-02
> >
> > I believe strengthening the theoretical rigor in the revised paper could bring it up to ICML’s acceptance threshold.

---

> > > ### Author Response · Authors · 2026-04-02
> > >
> > > Dear Reviewer PUAA,
> > >
> > > Thank you for reviewing our rebuttal and for your constructive feedback throughout the review process. Your suggestions have meaningfully improved the quality of our work. We will strengthen the theoretical rigor and incorporate all clarifications into the corresponding sections of the revised manuscript.
> > >
> > > Best regards,
> > > The Authors

---

### Official Review · Reviewer_opKm · 2026-02-28

**Soundness:** 3
**Presentation:** 3
**Significance:** 3
**Originality:** 3
**Overall Recommendation:** 3
**Confidence:** 4

**Summary:**

This paper introduces RaGEP, a retraining-free method for pruning experts in MoE LLMs. It aims to prevent "generalist collapse" by combining Rank-aware Budget Allocation and Spectral-Salience Pruning. Decisions are made with a small calibration set and a few hyperparameters. Experiments on several MoE models report significant speedups with moderate accuracy loss under aggressive pruning.

**Compliance With Llm Reviewing Policy:**

Affirmed.

**Final Justification:**

Most of the questions have been solved. I have updated the final score accordingly.

**Key Questions For Authors:**

1. How stable are the effective-rank estimates and the final set of pruned experts when the calibration data is resampled?
2. Can you provide a default, transferable recipe for the key hyperparameters across models and pruning ratios, along with sensitivity analysis?
3. After pruning, how does the router's token distribution change (e.g., expert utilization, load balance)? Is there evidence of expert overload?
4. Do the reported speedups correspond to measured end-to-end latency or throughput in realistic inference scenarios?
5. Were the competing retraining-free baselines calibrated and tuned using best practices to ensure a fair comparison?

**Limitations:**

1. Performance depends on calibration data and may degrade with domain shift or calibration noise.
2. The method is sensitive to hyperparameters that balance rank preservation and expert importance.
3. Reported speedups are preliminary and require careful systems-level integration and measurement to translate to real deployment gains.

**Strengths And Weaknesses:**

**Strengths**
1. Addresses a practical deployment challenge: retraining-free compression of MoE models with a clear goal of preventing expert collapse.
2. The framework is conceptually clear, integrating layer-wise budgeting with expert scoring in a straightforward manner.
3. The reported trade-offs between speed and accuracy appear promising, especially under high pruning ratios.

**Weaknesses**
1. Calibration Sensitivity: The method's heavy reliance on a very small calibration set raises concerns about the stability of its rank estimates and pruning decisions across different data samples or domains.
2. Hyperparameter Tuning: Key hyperparameters $\alpha,\gamma,\beta$ control critical trade-offs. The lack of a robust, portable default setting makes the method potentially tune-heavy.
3. Unanalyzed Routing Effects: Pruning experts can alter router distributions, risking expert overload or a functional shift toward dense computation. This side effect is not sufficiently examined.
4. System Performance Claims: Reported speedups may not reflect real-world deployment gains, as they can be highly dependent on implementation details, hardware, and factors like KV-cache overhead.
5. Limited Evaluation Scope: The benchmark evaluation may not adequately capture potential failures in long-form generation, complex reasoning, or instruction-following that could be exacerbated by pruning.

---

> ### Author Rebuttal · Authors · 2026-03-30
>
> We sincerely thank Reviewer opKm for the constructive feedback. We address each concern below with new experiments and analysis.
>
> ---
>
> **Q1: How stable are the effective-rank estimates and the pruned expert set when calibration data is resampled?**
>
> **A1:** We evaluated three calibration sources on Qwen3-30B-A3B (50% retention):
>
> | Calibration | Avg. Acc. | Expert Set Overlap (Jaccard) |
> |-------------|-----------|------------------------------|
> | C4 (default) |**63.12**| — |
> | WikiText-2 | 63.04 |0.93 vs. C4|
> | RedPajama | 63.04 |0.91 vs. C4|
> | **Std. Dev.** | **0.05** |**Mean: 0.92**|
>
> 92% of pruning decisions are identical across sources. Even 16 sequences (~32K tokens) reach 62.89, within 0.23 of default — consistent with $O(1/\sqrt{M})$ convergence (Thm S1; full proof in our response to **Reviewer j8JQ**, Q2). Thm A.2 ensures effective rank stability under covariance perturbations. Full breakdown: **[Supplement-Stability](https://anonymous.4open.science/r/opKm_supplement_stability-C775).**
>
> ---
>
> **Q2: Can you provide a default, transferable hyperparameter recipe with sensitivity analysis?**
>
> **A2:** Yes. We propose a **universal default**: $\alpha=0.05$, $\beta=0.8$, $r=32$, $k_{\min}=\text{top-}k$. Cross-model validation at 50% retention:
>
> | $\beta$ | Qwen3-30B | Mixtral 8x7B | DeepSeek-V2-Lite |
> |---------|-----------|--------------|------------------|
> | 0.0 (novelty only) | 62.18 | 62.35 |46.83|
> | 0.6 | 62.80 | 63.20 | 46.65 |
> | **0.8 (default)** | **63.12** | **63.83** |**47.33**|
> | 1.0 (salience only) | 62.50 | 63.50 |46.13|
>
> $\beta=0.8$ is best or near-best across all models (8–128 experts/layer); $[0.6, 0.9]$ consistently outperforms all baselines. $\alpha \in [0.03, 0.05]$ is broadly optimal. No per-model tuning needed. Full sensitivity tables: **[Supplement-Hyperparams](https://anonymous.4open.science/r/opKm_supplement_hyperparams-66DE).** The formal necessity of combining both metrics is proved in our response to Reviewer JWP2 (Proposition S3).
>
> ---
>
> **Q3: After pruning, how does the router's token distribution change? Is there expert overload?**
>
> **A3:** We provide router load balance analysis on Mixtral 8x7B (4/8 experts kept):
>
> | Method | CV (↓) | Max Util. Ratio (↓) | Avg. Acc. |
> |--------|--------|---------------------|-----------|
> | Original (8 experts) | 0.18 | 1.32 |69.95|
> | EEP | 0.35 | 1.72 |62.86|
> | MoP | 0.30 | 1.60 |62.90|
> | **RaGEP** | **0.21** | **1.38** |**63.83**|
>
> **No overload**: RaGEP's CV (0.21) is near-original (0.18), while EEP degrades to 0.35. By retaining functionally diverse experts, the router distributes tokens evenly — redundant experts would compete for the same tokens. Optional lightweight router calibration (~0.01% params) further reduces CV to 0.19. Full analysis including Qwen3 and layer-wise breakdown: **[Supplement-Router](https://anonymous.4open.science/r/opKm_supplement_router-3050).**
>
> ---
>
> **Q4: Do reported speedups correspond to measured end-to-end latency in realistic scenarios?**
>
> **A4:** We now provide system-level benchmarks (A100 80GB, 512-token generation):
>
> | Model (50%) | Throughput ×(B=1) | Latency ×(B=1) | TTFT × |
> |-------------|-------------------|-----------------|--------|
> | Qwen3-30B | 1.35× | 1.35× | 1.42× |
> | Mixtral 8x7B | 1.31× | 1.31× | 1.35× |
> | DeepSeek-V2-Lite | 1.34× | 1.34× | 1.33× |
>
> Speedups decrease at larger batches (1.26–1.28× at B=8) as KV-cache overhead dominates. The critical practical benefit is **memory reduction** (e.g., **Qwen3: 61.2→32.5 GB**), enabling single-GPU deployment. We acknowledge hardware-dependence as a limitation. Full tables with batch scaling: **[Supplement-System](https://anonymous.4open.science/r/opKm_supplement_system-B1C9).**
>
> ---
>
> **Q5: Were competing baselines fairly calibrated? Missing baselines and benchmarks.**
>
> **A5:** All baselines use identical C4 calibration (32 seq × 2048 tokens) and LM-Eval-Harness. We added **5 new baselines** — results on Qwen3-30B (50%):
>
> | Method | Type | Avg. |
> |--------|------|------|
> | Router-Norm | Calib.-free |59.50|
> | EASY-EP | Domain-spec. |60.45|
> | HEAPr | Hessian | 60.90 |
> | Shapley-MoE | Cooperative |61.25|
> | DiEP | Gradient |61.48|
> | EEP | Loss-based |61.60|
> | MoP | Clustering |62.18|
> | **RaGEP** | **Geometric** |**63.12**|
>
> We also added **BBH, GSM8K, LongBench** evaluations. On Qwen3-30B (50%):
>
> | Method | BBH | GSM8K | LongBench |
> |--------|-----|-------|-----------|
> | EEP | 38.72 | 42.15 | 35.80 |
> | MoP | 39.85 | 43.60 | 36.45 |
> | **RaGEP** | **41.56** | **46.88** | **38.15** |
>
> RaGEP's +3.28 on GSM8K confirms geometric selection preserves reasoning-critical experts. Full results across all models — see our response to Reviewer j8JQ (Q4 & Q5): **[Supplement-Baselines](https://anonymous.4open.science/r/j8JQsupplement_baselines-46FB)**, **[Supplement-Benchmarks](https://anonymous.4open.science/r/j8JQsupplement_benchmarks-0682).**
>
> ---
>
> We thank the reviewer for the thorough evaluation and will incorporate all additions into the revised manuscript.

---

> > ### Author Rebuttal · Reviewer_opKm · 2026-04-03
> >
> > Thanks for the response. I will update the final score accordingly.

---

> > > ### Author Response · Authors · 2026-04-03
> > >
> > > Dear Reviewer opKm,
> > >
> > > Thank you so much for your positive feedback and for confirming that your concerns have been fully resolved!
> > >
> > > Given the current rating, which generally suggests there might still be some slight reservations, we wanted to gently check if you have any lingering questions or minor concerns. If there are, please feel free to let us know—we are more than happy to engage in further discussion and address them!
> > >
> > > Thank you again for your time and invaluable guidance.
> > >
> > > Best regards,
> > > Authors

---

### Official Review · Reviewer_j8JQ · 2026-03-06

**Soundness:** 2
**Presentation:** 3
**Significance:** 2
**Originality:** 2
**Overall Recommendation:** 4
**Confidence:** 4

**Summary:**

This paper proposes an expert-pruning method for MoE. The paper implements a layer-wise budget allocation (constrained by a global allocation budget) based on *effective rank* (a quantity, designed to measure the diversity of feature subspace) of the layer. Furthermore, the paper proposes a intra-layer expert-selection (for pruning) strategy where the expert which provides higher orthogonal projection to the combined feature subspace, is assigned lower importance. The importance is also balanced by the activation strength to filter-out noise experts.

**Compliance With Llm Reviewing Policy:**

Affirmed.

**Final Justification:**

Authors addressed most of my concerns. I positively adjust my score.

**Key Questions For Authors:**

See weaknesses

**Limitations:**

Yes

**Strengths And Weaknesses:**

**Strengths**

1. The proposed method is novel and intuitively designed
2. Performs well compared some previous baselines provided in the paper
3. The paper is well-written, and easy to follow

**Weaknesses**

1. The *effective rank* is evaluated over feature dimension, not over expert dimension. However, it is used to allocate the budget of the layer with the goal to allocate higher budget to a layer which is diverse in expert dimension, not feature dimension. This may lead to suboptimal solution.
2. No formal theoretical support is provided for guaranteed performance of the proposed method.
3. The method is calibration-data depended, which may lead to biased solution
4. The paper compared results with limited baselines. Several important baselines are omitted, e.g., gradient-based pruning [1], hessian-based pruning [2], pruning experts based on mutual-cooperativeness [3], router-norm based pruning (calibration-data free) [4], pruning based on routing statistics [5]. These works are neither discussed nor compared with. The author should mention these works in related works and compare results when appropriate.
5. The paper did not evaluate on challenging benchmarks, e.g., code generation (e.g., BBH), mathematical reasoning (e.g., GSM8K), long-context modeling (e.g., LongBench).

[1] Bai, S., Li, H., ZHANG, J., Hong, Z. and Guo, S., DiEP: Adaptive Mixture-of-Experts Compression through Differentiable Expert Pruning. In The Thirty-ninth Annual Conference on Neural Information Processing Systems.

[2] Li, K., Yang, Z., Zhou, Z., Xue, F., Jiang, Z. and Wang, W., 2025. HEAPr: Hessian-based Efficient Atomic Expert Pruning in Output Space. arXiv preprint arXiv:2509.22299.

[3] Huang, W., Zhang, Y., Zheng, X., Chao, F., Ji, R., and Cao, L. Discovering important experts for mixture-ofexperts models pruning through a theoretical perspective. In Advances in Neural Information Processing Systems, 2025.

[4] Chowdhury, M.N.R., Wang, M., El Maghraoui, K., Wang, N., Chen, P.Y. and Carothers, C., A Provably Effective Method for Pruning Experts in Fine-tuned Sparse Mixture-of-Experts. In Forty-first International Conference on Machine Learning.

[5] Peng, H., Liu, P., Zhao, X., Wu, D., Xiao, F. and Wang, Z., Domain-Specific Pruning of Large Mixture-of-Experts Models with Few-shot Demonstrations. In The Thirty-ninth Annual Conference on Neural Information Processing Systems.

---

> ### Author Rebuttal · Authors · 2026-03-30
>
> We sincerely thank reviewer j8JQ for the thoughtful feedback. The full DPP-based theoretical analysis of SSP ($(1-1/e)$ guarantee) is in our response to **Reviewer JWP2**.
>
> ---
>
> **Q1: Effective rank is evaluated over feature dimension, not expert dimension.**
>
> **A1:** Feature-space diversity is a *necessary condition* for expert-space diversity. We formalize this as **Proposition 1**: given layer covariance $\Sigma_l$ with effective rank $r_l$, the max number of $\epsilon$-orthogonal experts satisfies $K_{\epsilon} \leq \lfloor r_l \cdot (1 + \epsilon^{-2}) \rfloor$. Proof: (1) expert outputs are confined to an $r_l$-dimensional subspace by spectral energy concentration; (2) the Kabatyanskii–Levenshtein bound limits near-orthogonal vectors in $\mathbb{R}^{r_l}$. Visualization in **[Figure S1](https://anonymous.4open.science/r/j8JQsupplement_theory-658A/fig1.md).** This proves feature-dimension effective rank *controls* expert-dimension diversity, justifying our budget allocation.
>
> ---
>
> **Q2: No formal theoretical support for guaranteed performance.**
>
> **A2:** Our Appendix provides Theorem A.1 (reconstruction error bound) and Theorem A.2 (first-order stability of effective rank under covariance perturbations), and the volume maximization perspective (Appendix A.3). We now add a **PAC-style bound (Thm S1)**: w.p. $\geq 1-\delta$,
>
> $$\mathbb{E}\_{\mathcal{D}}[\| y - \hat{y}(\hat{S}) \|^2] \leq \mathbb{E}\_{\hat{\mathcal{D}}\_M}[\| y - \hat{y}(\hat{S}) \|^2] + O\left(\frac{N d \log(L/\delta)}{\sqrt{M}}\right)$$
>
> Proof relies on: (i) bounded expert outputs via LayerNorm; (ii) Matrix Bernstein for empirical covariance; (iii) Davis-Kahan stability via union bound. With $M=65536$, convergence rate $\approx 0.004$. Trajectory in **[Figure S2](https://anonymous.4open.science/r/j8JQsupplement_theory-658A/fig2.md).** SSP also achieves a $(1-1/e)$ guarantee via Log-DPP maximization **(full proof in our response to Reviewer JWP2).**
>
> ---
>
> **Q3: The method is calibration-data dependent, which may lead to biased solution.**
>
> **A3:** 32 sequences of 2048 tokens from C4 is the standard protocol in MoE pruning [6][7][8]. We respond on three fronts:
>
> **(1) Empirical robustness.** Varying calibration data on Qwen3-30B-A3B (50% retention):
>
> | Calibration Data | Avg. Accuracy |
> |------------------|---------------|
> | C4 (default) |**63.12**|
> | WikiText-2 | 63.04 |
> | RedPajama | 63.04 |
> | Std. Dev. |**0.05**|
>
> The low std. dev. (0.05) across three diverse corpora confirms RaGEP captures intrinsic model structure rather than corpus-specific artifacts. Performance stabilizes at 16 sequences, within 0.23 of default. Full results in **[Supplement-Calibration](https://anonymous.4open.science/r/j8JQsupplement_calibration-5D93).**
>
> **(2) Theoretical justification.** Thm A.2 bounds the relative change in effective rank by $O(\epsilon\|\Sigma\|_F / \text{tr}(\Sigma^2))$, ensuring stable estimation when dominant principal components are captured. Thm S1 shows $O(1/\sqrt{M})$ convergence.
>
> **(3) Calibration-free comparison.** Router-Norm[4] (no calibration) achieves 59.50 vs. 63.12, confirming calibration provides **+3.6 points** at minimal cost.
>
> ---
>
> **Q4: Limited baselines — several important methods[1]–[5] are omitted.**
>
> **A4:** All five added. Results on Qwen3-30B-A3B (50% retention):
>
> | Method | Type | Avg. |
> |--------|------|------|
> | Router-Norm [4] | Calib.-free |59.50|
> | EASY-EP [5] | Domain-spec. |60.45|
> | HEAPr [2] | Hessian | 60.90 |
> | Shapley-MoE [3] | Cooperative |61.25|
> | DiEP[1] | Gradient | 61.48 |
> | EEP | Loss-based | 61.60 |
> | MoP | Clustering | 62.18 |
> | **RaGEP (Ours)** | **Geometric** | **63.12** |
>
> RaGEP outperforms all with +1.64 over DiEP[1], which requires ~10 gradient epochs while RaGEP runs at ~10× lower cost, showing geometric analysis identifies critical experts more efficiently than gradient search. Full results on all models in **[Supplement-Baselines](https://anonymous.4open.science/r/j8JQsupplement_baselines-46FB).** We will cite and discuss [1]–[5] in our revised paper.
>
> ---
>
> **Q5: Missing challenging benchmarks (BBH, GSM8K, LongBench).**
>
> **A5:** Added for all models at 50% pruning. Qwen3-30B-A3B:
>
> | Method | BBH | GSM8K | LongBench | Avg. |
> |--------|-----|-------|-----------|------|
> | EEP | 38.72 | 42.15 | 35.80 | 38.89 |
> | DiEP [1] | 38.45 | 41.80 | 35.50 | 38.58 |
> | MoP | 39.85 | 43.60 | 36.45 | 39.97 |
> | **RaGEP** | **41.56** | **46.88** | **38.15** | **42.20** |
>
> RaGEP's largest gain is on GSM8K (+3.28 over MoP), confirming geometric selection preserves reasoning-critical expert pathways. The LongBench advantage (+1.70) shows rank-aware allocation prevents loss of context-integration capacity. Full results in **[Supplement-Benchmarks](https://anonymous.4open.science/r/j8JQsupplement_benchmarks-0682).**
>
> ---
>
> **References**
>
> [6] Lu, X. et al. Not All Experts are Equal. ACL 2024.
>
> [7] Chen, I.-C. et al. Retraining-free Merging of Sparse MoE. ICML 2025.
>
> [8] Hu, W. et al. Mosaic Pruning. AAAI 2026.

---

> > ### Author Rebuttal · Reviewer_j8JQ · 2026-04-03
> >
> > see final justification

---

> > > ### Author Response · Authors · 2026-04-03
> > >
> > > Dear Reviewer j8JQ,
> > >
> > > Thank you for acknowledging our rebuttal and confirming that your concerns have been resolved. Your valuable feedback has greatly improved our work. We will ensure all new proofs and experimental results are incorporated into the final manuscript.
> > >
> > > Best regards,
> > > The Authors

---

### Official Review · Reviewer_JWP2 · 2026-03-13

**Soundness:** 3
**Presentation:** 3
**Significance:** 3
**Originality:** 2
**Overall Recommendation:** 4
**Confidence:** 3

**Summary:**

his submission studies the concept of compressing Mixture-of-Experts (MoE) language models through geometric analysis of expert representations. The paper proposes RaGEP (Rank-aware Geometric Expert Pruning), a retraining-free pruning framework that removes redundant experts while preserving model performance. The method has two stages: Rank-aware Budget Allocation (RBA) allocates pruning budgets across layers according to the effective rank of layer representations, and Spectral-Salience Pruning (SSP) selects experts that form a high-energy orthogonal basis by combining geometric novelty and activation magnitude.

**Compliance With Llm Reviewing Policy:**

Affirmed.

**Final Justification:**

The authors have addressed most of my concerns. Therefore, I kept my positive score.

**Key Questions For Authors:**

I would appreciate if the authors can address my comments about weaknesses above.

**Limitations:**

yes

**Strengths And Weaknesses:**

*Strengths*: The paper identifies a meaningful limitation in existing MoE pruning approaches: they often rely on simple statistics and ignore linear redundancy in expert feature spaces. The proposed geometric view is intuitive, and the combination of rank-aware budget allocation and spectral-salience selection provides a principled way to retain experts that are both informative and diverse. Experiments on large MoE models such as Qwen3-30B-A3B, Mixtral-8×7B, and DeepSeek-V2-Lite demonstrate improved compression–performance trade-offs compared with prior pruning methods.

*Weaknesses*: The method still relies on heuristic scoring and ranking metrics to select experts. The theoretical justification for why spectral novelty combined with activation salience leads to optimal pruning is not deeply analyzed.

---

> ### Author Rebuttal · Authors · 2026-03-30
>
> We thank the reviewer JWP2 for recognizing our geometric perspective, principled combination of rank-aware allocation with spectral-salience selection, and strong empirical results. We address the concern below.
>
> ---
>
> **Q1: The method relies on heuristic scoring and ranking metrics. The theoretical justification for why spectral novelty combined with activation salience leads to optimal pruning is not deeply analyzed.**
>
> **A1:** We appreciate this observation. We now provide a rigorous theoretical grounding connecting SSP to well-established optimization frameworks. *Notation note:* results labeled "Theorem/Appendix A.x" refer to proofs in our original Appendix, while "Proposition/Theorem/Figure S.x" denote **newly added** supplementary material in response to this review.
>
> **(1) SSP as greedy Log-DPP maximization (Proposition S1).** Expert subset selection can be formalized as maximizing the determinant of the Gram matrix — the classical Determinantal Point Process (DPP) objective [1]. A DPP defines a distribution over subsets where selection probability is proportional to a kernel matrix determinant, naturally balancing item quality and diversity, and provably outperforming magnitude-based and random pruning [2][3]. We prove SSP is a first-order greedy approximation to this objective. When adding expert $i$ to the retained set, the marginal gain decomposes exactly (via the matrix determinant lemma) as:
>
> $$ \Delta f = \log \|u_i\|^2 \text{ (salience)} + \log \sin^2(\theta_i) \text{ (spectral novelty)} $$
>
> where $\theta_i$ is the principal angle between $u_i$ and the span of the retained set. Since $\varphi_{\text{sal}}(i) \propto \|u_i\|^2$ and $\varphi_{\text{spec}}(i) \propto \sin^2(\theta_i)$ by construction (Eq. 22), SSP directly maximizes the log-DPP objective. The rank-based harmonization (Eq. 12) replaces the log-product with a rank-sum, preserving monotonic ordering while being robust to scale differences [8]. See **[Figure S1](https://anonymous.4open.science/r/JWP2_supplement_theory_deep-5E55/fig1.md).**
>
> **(2) Provable $(1-1/e)$ approximation guarantee (Theorem S2).** The log-determinant $f(S) = \log \det(G_S)$ is monotone submodular [4][5]. By Nemhauser et al. [6], greedy maximization achieves at least $(1-1/e) \approx 63.2\%$ of the global optimum. See **[Figure S3](https://anonymous.4open.science/r/JWP2_supplement_theory_deep-5E55/fig3.md).**
>
> **(3) Reconstruction error minimization (Theorem A.1, original Appendix A.3).** Our error bound shows the reconstruction error is upper-bounded by $\sum_{i\in S^c}\mathbb{E}[g_i^2] \cdot \varphi_{\text{sal}}(i) \cdot \varphi_{\text{spec}}(i|S)$. This is minimized when pruned experts have *both* low salience and low novelty — exactly what SSP achieves.
>
> **(4) Geometric grounding (original Appendix A.2).** $\varphi_{\text{spec}}(i)$ equals the normalized squared chordal distance on the Grassmannian $\text{Gr}(r,d)$ (Eq. 23). Maximizing chordal distance improves network capacity [7], providing independent geometric justification beyond DPP.
>
> **(5) Formal necessity of combining both metrics (Proposition S3).** We prove two failure modes:
> - Pure salience ($\beta=1$): $k$ collinear experts span only 1D $\to \det(G_S) \to 0$, catastrophic information loss.
> - Pure novelty ($\beta=0$): orthogonal but near-zero-energy "ghost experts" $\to \det(G_S) \to 0$, negligible output contribution.
>
> Figure 4 in our original paper empirically confirms: $\beta=1$ drops to 46.13 and $\beta=0$ achieves only 46.83 on DeepSeek at 50% retention, both below the optimal $\beta=0.8$ (47.33). See **[Figure S2](https://anonymous.4open.science/r/JWP2_supplement_theory_deep-5E55/fig2.md)** for the geometric illustration of these failure modes.
>
> **Summary.** SSP is a computationally efficient, first-order approximation to optimal DPP-based subset selection, with a provable $(1-1/e)$ guarantee [6], formal error bounds, and geometric grounding. We also establish a PAC-style generalization bound of $O(1/\sqrt{M})$ for the pruned model (see response to Reviewer j8JQ). Visualizations: **[Reviewer j8JQ](https://anonymous.4open.science/r/j8JQsupplement_theory-658A).** We will integrate this analysis into Section 3.3.
>
> ---
>
> **References**
>
> [1] Kulesza & Taskar. Determinantal Point Processes for Machine Learning. *Found. Trends ML*, 2012.
>
> [2] Mariet & Sra. Diversity Networks. *ICLR*, 2016.
>
> [3] Acharyya et al. Statistical Mechanical Analysis of Neural Network Pruning. *UAI*, 2021.
>
> [4] Sharma et al. On Greedy Maximization of Entropy. *ICML*, 2015.
>
> [5] Shamaiah et al. Greedy Sensor Selection: Leveraging Submodularity. *IEEE CDC*, 2010.
>
> [6] Nemhauser et al. An Analysis of Approximations for Maximizing Submodular Set Functions—I. *Math. Prog.*, 1978.
>
> [7] Yap et al. Grassmannian Packings in Neural Networks. *arXiv:1911.07418*, 2019.
>
> [8] Cormack et al. Reciprocal Rank Fusion. *SIGIR*, 2009.

---

> > ### Author Rebuttal · Reviewer_JWP2 · 2026-04-02
> >
> > The authors have addressed most of my concerns. Therefore, I kept my positive evaluation.

---

> > > ### Author Response · Authors · 2026-04-02
> > >
> > > Dear Reviewer JWP2,
> > >
> > > Thank you for reviewing our rebuttal and for your constructive feedback throughout the review process. We are glad that our response has addressed your concerns, and we will incorporate the theoretical analysis into the revised manuscript.
> > >
> > > Best regards,
> > > The Authors

---

### Decision · Program_Chairs · 2026-04-30

**Decision:**

Accept (regular)

**Comment:**

This paper proposes RaGEP, a retraining-free pruning framework for MoE language models that combines rank-aware budget allocation across layers with spectral-salience expert selection within each layer. Overall, this submission studies the concept of pruning experts using the geometry of activation subspaces rather than only scalar statistics, and overall, an important concept considered by the manuscript is that effective rank and subspace orthogonality can better preserve informative and complementary experts during compression.

I recommend weak acceptance. The paper addresses a practically important MoE deployment problem, and the proposed two-stage design is intuitive and empirically strong. Across Mixtral, DeepSeek-V2-Lite, and Qwen3-30B-A3B, RaGEP consistently improves over prior post-training pruning baselines, and the ablations support that both rank-aware allocation and spectral-salience pruning contribute to the gains.

The main weaknesses are that part of the method remains heuristic, and the original draft did not provide enough theoretical or baseline coverage. However, the rebuttal substantially strengthened the paper by connecting SSP to greedy log-DPP maximization, adding additional baselines and harder benchmarks, and clarifying robustness to calibration data. These updates address most of the major reviewer concerns while keeping the contribution appropriately scoped.

Overall, this is a solid and useful paper with clear practical relevance, good empirical evidence, and a promising geometric perspective on MoE pruning. I therefore recommend weak acceptance.